# Time-resolved β-lactam cleavage by L1 metallo-β-lactamase

M. Wilamowski [1,2,3], D. A. Sherrell[4], Y. Kim [1,4], A. Lavens [4], R. W. Henning[5], K. Lazarski[4], A. Shigemoto[6], M. Endres[1], N. Maltseva [1], G. Babnigg [1], S. C. Burdette [6], V. Srajer[5] & A. Joachimiak [1,2,4] ✉

Serial x-ray crystallography can uncover binding events, and subsequent chemical conversions occurring during enzymatic reaction. Here, we reveal the structure, binding and cleavage of moxalactam antibiotic bound to L1 metallo-β-lactamase (MBL) from *Stenotrophomonas maltophilia*. Using time-resolved serial synchrotron crystallography, we show the time course of β-lactam hydrolysis and determine ten snapshots (20, 40, 60, 80, 100, 150, 300, 500, 2000 and 4000 ms) at 2.20 Å resolution. The reaction is initiated by laser pulse releasing $Zn^{2+}$ ions from a UV-labile photocage. Two metal ions bind to the active site, followed by binding of moxalactam and the intact β-lactam ring is observed for 100 ms after photolysis. Cleavage of β-lactam is detected at 150 ms and the ligand is significantly displaced. The reaction product adjusts its conformation reaching steady state at 2000 ms corresponding to the relaxed state of the enzyme. Only small changes are observed in the positions of $Zn^{2+}$ ions and the active site residues. Mechanistic details captured here can be generalized to other MBLs.

Treatment of bacterial infections with β-lactam based antibiotics is the most common approach used in clinics and currently accounts for ~65% of all applied antibiotics. These widely used compounds include drugs with a broad-spectrum substrate profile such as penicillins, cephalosporins, monobactams, and carbapenems. Antibiotics with β-lactam core mimic substrates of bacterial transpeptidases essential for cell-wall remodeling and they block the formation of cross-bridges between adjacent peptidoglycan chains causing bacteria death. The massive world-wide use of β-lactam antibiotics resulted in microorganisms becoming drug resistant and this represents a major threat to human health and wellbeing[1,2]. Many multidrug-resistant (MDR) and extensively drug-resistant (XDR) microbial strains require elaborate treatments or escape available cures entirely. It is projected that by 2050, the increase in mortality caused by infections with MDR/XDR pathogens will grow beyond the level predicted for cancer[3]. New

treatments and discovery of more effective inhibitors for enzymes responsible for MDR/XDR are urgently needed.

The most important mechanism of β-lactam antibiotics resistance is enzymatic hydrolysis by the bacterial β-lactamases. The β-lactamase's genes are of natural origin, and are broadly present in various environments, including those that have not been exposed to synthetic antibiotics. The genes are often located on mobile plasmids and can integrate with and be maintained on the bacterial chromosome[4]. The enzymes are frequently very promiscuous, but also evolve by mutations in response to the presence of new antibiotics. There are four classes of β-lactamases (A, B, C and D)[4]. The most important enzymes belong to class B and bind one or two metal ions, typically $Zn^{2+}$, playing key roles in the substrate binding and catalysis. They are called metallo-β-lactamases (MBL) and are divided into three subclasses based on their protein sequence, substrate profiles and

[1]Center for Structural Genomics of Infectious Diseases, Consortium for Advanced Science and Engineering, University of Chicago, Chicago, IL 60667, USA. [2]Department of Biochemistry and Molecular Biology, University of Chicago, Chicago, IL 60637, USA. [3]Department of General Biochemistry, Faculty of Biochemistry, Biophysics and Biotechnology of Jagiellonian University, 30387 Krakow, Poland. [4]Structural Biology Center, X-ray Science Division, Argonne National Laboratory, Argonne, IL 60439, USA. [5]Center for Advanced Radiation Sources, University of Chicago, Chicago, IL 60637, USA. [6]Department of Chemistry and Biochemistry, Worcester Polytechnic Institute, Worcester, MA 01609, USA. ✉e-mail: andrzejj@anl.gov

variations in their active site configurations for $Zn^{2+}$ ions. MBLs from B1 and B3 subclasses are highly promiscuous and can hydrolyze nearly every β-lactam containing antibiotic, including the most recently developed last-resort carbapenems. Substrate promiscuity of MBLs stems from their structure having a large, open binding site consisting of multiple flexible loops and a di-nuclear $Zn^{2+}$ scaffold which anchors β-lactam moiety of a broad range of substrates carrying different substituents. In contrast, class B2 enzymes have a narrow spectrum carbapenem substrate profile, with poor activity toward penicillins and cephalosporins. Notably, MBLs are not inactivated by mechanism-based inhibitors like FDA-approved clavulanate or tazobactam that form an irreversible covalent adduct with the enzyme[5]. Studies of MBLs to develop new treatments and prevent another possible global pandemic yielded, thus far, no breakthrough [6,7].

One of the emerging MBLs is L1 from *Stenotrophomonas maltophilia*, belonging to the B3 subclass of MBLs[8]. The *S. maltophilia* is an opportunistic Gram-negative bacillus pathogen found in various aqueous habitats that has become MDR. It causes infections, mainly among hospitalized patients, and has been associated with high morbidity and mortality in severely immunocompromised and debilitated individuals treated for cancer, cystic fibrosis, and transplants. The L1 MBL gene is integrated with the *S. maltophilia* chromosome. Recently we have reported several high-resolution crystal structures of this promiscuous enzyme, including the metal-free apo form, complexes with bi-metal ion scaffolds ($Zn^{2+}$, $Cd^{2+}$ and $Cu^{2+}$), a complex with captopril inhibitor and several complexes with hydrolyzed antibiotics (imipenem, moxalactam, meropenem, and penicillin G)[9]. L1 MBL shares close structural similarity of bi-metal ions architecture of the catalytic site with the NDM-1 from *K. pneumoniae*[6,9,10] and other MBLs.

The catalytic mechanism for MBLs has been studied extensively[4,6–8,10–19] and recently reviewed by Bahr et al.[20]. The MBLs use a mono- or bi-metal scaffold to direct the hydrolysis of the β-lactam ring[7,8]. Substrates for MBLs include penicillin G, present in environment, or synthetic carbapenems developed by pharmaceutical industry such as imipenem, moxalactam, and meropenem[15,21]. The proposed mechanism includes a nucleophilic water molecule, either $Zn^{2+}$ ions bridging water or hydroxide or activated water originated from bulk solvent attacking the carbonyl carbon of the β-lactam ring[22]. The $Zn^{+2}$ ions are acting as a Lewis acid by coordinating the β-lactam carbonyl oxygen and facilitating nucleophilic attack. The positive charge of metal ions offsets the negative charge developed on the oxygen atom in the proposed tetrahedral intermediate anions. Alternative hypotheses supported by quantum mechanics/molecular mechanics computations proposed that the bridging water bound to $Zn^{2+}$ atoms is not a nucleophile, and the bulk solvent water molecule is the nucleophile mandatory for the reaction[23]. Proposed MBLs mechanisms indicate that H118 together with D120 may act as a general base coordinating the water molecule activated by zinc center to form hydroxide which initiates nucleophilic attack on the carbonyl carbon of β-lactam opening the ring. All these proposed reaction steps have been modelled computationally and some were confirmed experimentally using biochemistry, spectroscopic trapping of intermediates using electron paramagnetic resonance, Raman spectroscopies, extended x-ray absorption fine structure (EXAFS) and x-ray absorption near edge structure (XANES) spectroscopies[11,16,23–27]. Structurally, only complexes with hydrolyzed or partly hydrolyzed antibiotics have been available thus far[6,9,10]. The continuous debate about mechanisms of MBLs action was the impulse to perform time-resolved (TR) studies to determine structures of MBL in complex with substrate before the reaction occurs and to capture intermediates.

The time course and corresponding binding and chemical transformations during enzyme catalysis were concealed for many years. TR crystallography, traditionally done using room-temperature synchrotron data from a single crystal, can reveal structures of intermediates and provide critical information about mechanisms[28,29]. The molecular

events inside biological macromolecule crystals were studied using polychromatic beam and Laue diffraction with photoactive macromolecules, or using photolabile caged compounds[29–31]. Serial crystallography at free-electron lasers (XFELs) allows for collecting x-ray diffraction data from hundreds of thousands of crystals at room-temperature and determination of protein structures with reduced radiation dose[32,33]. TR serial femtosecond crystallography (TR-SFX) extended the time resolution from 100 ps available at light sources to femtoseconds and was used to visualize fast chemical events[34,35] for photosystem II, photoactive yellow protein, bacteriorhodopsin, and myoglobin[29,35,36]. The mechanism of β-lactamases involved in antibiotic resistance was studied using TR-SFX by Olmos et al.[37]. The BlaC lactamase from *Mycobacterium tuberculosis* with the serine in the active site was analyzed using the TR-SFX method at an XFEL using a mix-and-inject approach and the progress of the reaction was recorded from 5 to 2000 ms[36–38]. More recently, serial crystallography has also been successfully applied at synchrotron light sources (SSX)[33,39,40], including TR-SSX[41–43], both with monochromatic and polychromatic x-ray beams. The use of "pink" beam provides several advantages: increased x-ray photon flux and significant reduction the number of diffraction images needed for structure determination[29,41]. Therefore, "pink" beam SSX is a desirable technique in the case of analysis of enzyme mechanisms with multiple time-points. These technological advances lead us to the analysis of carbapenem resistant L1 MBL with polychromatic beam and TR-SSX at 14-ID beamline at the Advanced Photon Source (APS). In this work we report multiple structures of L1 MBL from *S. maltophilia* using room temperature SSX and TR-SSX and visualize the time course hydrolysis of moxalactam antibiotic. The moxalactam cleavage reaction occurs when laser pulse causes release of $Zn^{2+}$ ions from a UV-labile zinc-photocage in the presence of substrate. The experiments carried out over 4000 ms and the resulting high-resolution structures reveal $Zn^{2+}$ ions and substrate binding, β-lactam ring cleavage and ligand conformational adjustments in the active site.

## Results and discussion

Structural visualization of ligand chemical transformations during enzyme catalysis presents significant challenges on both time and length scales. These include effective and precise control of initiating the enzymatic reaction, tracking the reaction progress at appropriate time slicing at high resolution, structure refinement and analysis of experimental electron density and interpretation in the context of amassed published data. Recent advances in structural biology, such as room temperature SSX, provide opportunity to uncover details of biding events, and subsequent chemical transformations occurring during enzymatic reaction. Here we describe TR-SSX studies of L1 MBL from *S. maltophilia* and application of [Zn(XDPAdeCage)]$^+$ − a UV-photolabile compound to control initiation of reaction. The cage strongly coordinates $Zn^{2+}$ ion, which can be released nearly irreversibly via decarboxylation after irradiation with 347 nm UV light[44]. Released zinc binds to the active site of L1 MBL in solution and triggers cleavage of meropenem, which absorption spectrum changes after cleavage of β-lactam ring (Supplementary Fig. 1). To study reaction of β-lactam antibiotic cleavage by L1 MBL we used moxalactam due to high quality of electron density maps countered around ligand in L1 crystals[9]. For the TR-SSX experiments the [Zn(XDPAdeCage)]$^+$ and moxalactam substrate were presoaked in protein crystals. These crystals produced "dark" data set (Supplementary Table 1). Observed crystal packing reveals that active sites of L1 MBL are faced toward exposed solvent channels that possess a diameter of 55 Å (Supplementary Fig. 2). High − 55.6% solvent content and open "honeycomb" architecture of L1 MBL crystals facilitates diffusion of compounds used for the soaking of crystals before initiation of TR reaction. We assume that both [Zn(XDPAdeCage)]+ caged compound and moxalactam (13.5 Å at longest intraatomic distances) occupy crystal solvent channels before

reaction. After de-caging with UV light zinc ions rapidly diffuse and bind to the enzyme active site, enable substrate binding, and trigger the cleavage of β-lactam ring (see below).

The L1 MBL was selected for TR-SSX experiments because the enzyme represents excellent model system for MBLs involved in antibiotic resistance. It was originally proposed by Spencer[6] that because L1 hydrolyzes moxalactam with a small Km and a small kcat, it may allow trapping initial steps of the reaction. Moreover, the enzyme can be purified in milligram quantities and produce well diffracting crystals[9]. These high symmetry crystals ($P6_422$) reduce the number of diffraction images needed to solve and refine a structure and contain only one enzyme monomer in the asymmetric unit, simplifying the analysis. We collected all data using the fixed-target ALEX chip set up[45,46] (Fig. 1). On average, for each time point scanning, only three chips were needed for a complete data set. The L1 microcrystals

diffracted to better than 2.20 Å resolution (Supplementary Fig. 3a, b, c, Supplementary Table 1). Overall, complete data sets for apo-enzyme – "no zinc" (EDTA treated crystals), "one zinc" (EDTA treated crystals for a short time), "dark" (EDTA treated crystals with [Zn(XDPAdeCage)]+ and moxalactam added, no laser exposure), and the ten-time points were collected (Supplementary Table 1). The "pink" reference structure of L1 MBL (PDB entry 7L91) was determined by MR using the 1.60 Å resolution structure of L1 in a complex with hydrolyzed moxalactam (PDB entry 6UAC) as a search model. These structures were refined with good crystallographic statistics (Supplementary Table 2). In the single crystal experiments, we averaged electron density over all unit cells exposed to x-rays. In the SSX experiments we averaged electron density of over thousands of microcrystals. These data can reveal the time evolution of populations and may represent multiple states. Overall, all data sets produced high quality structures with good

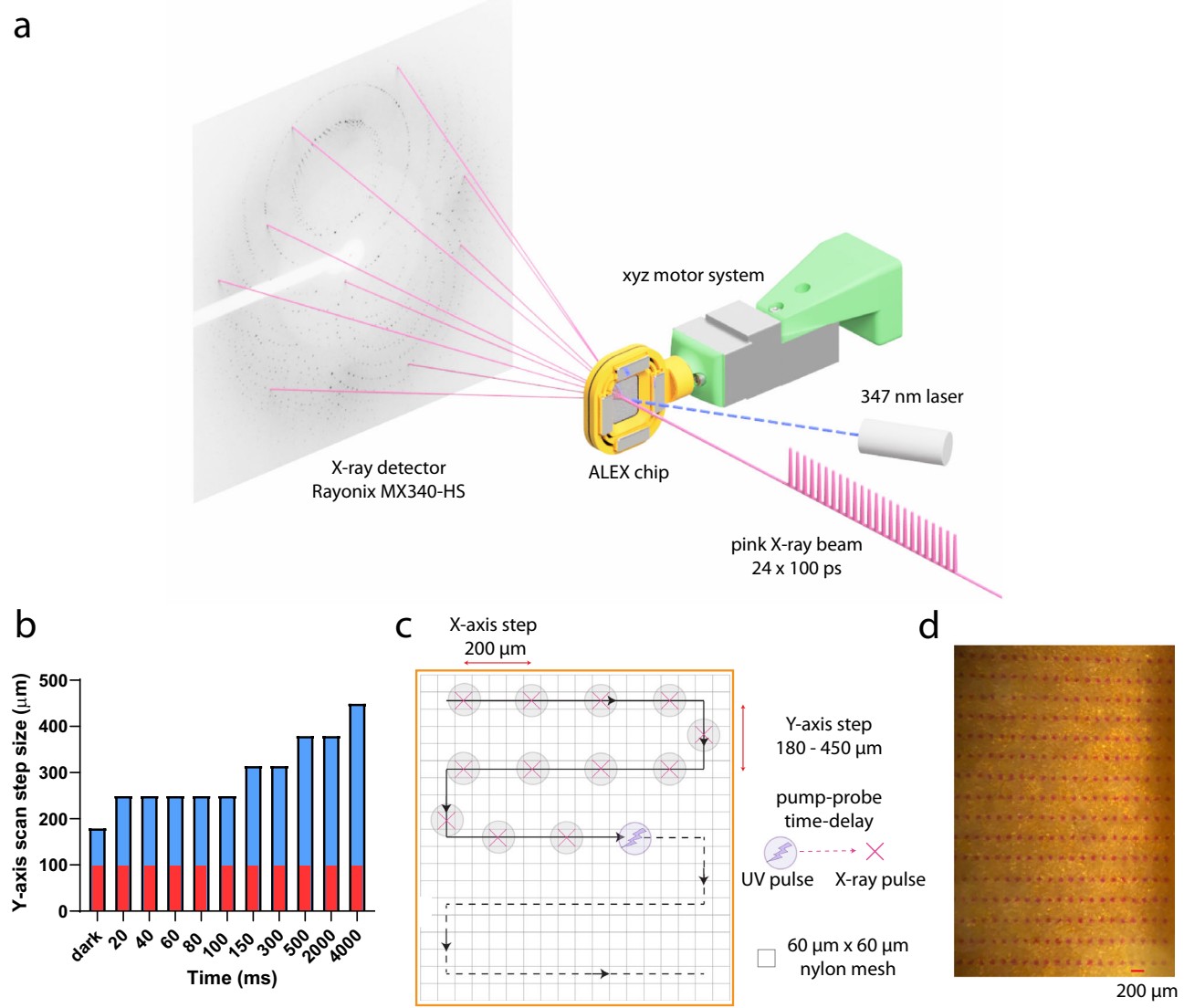

**Fig. 1 | Experimental set-up for TR-SSX at 14-ID-B BioCARS beamline.**
**a** Polychromatic x-ray beam depicted with simplified representation of pulse train exposure, containing 24 x-ray pulses with duration of 100 ps each and total pulse train duration of 3.7 μs. *ALEX* chip moves during scan in a meander-pattern over the preset area of the chip containing slurry of immobilized microcrystals. Laser pulse is coordinated with release of x-ray beam by a time delay that corresponds to a specific time point of reaction. The typical Laue diffraction pattern is shown obtained from L1 MBL crystal during TR-SSX experiment. **b** Movement of the *ALEX* chip across the *Y*-axis with step size labeled as blue bars with an overlaid red

baseline that indicates the diameter of a laser pulse spot, source data is available as Source data file. **c** Scheme of fixed-target SSX data collection on nylon mesh crystals holder utilizing the pump-probe system. The *X* and *Y* axis step sizes of the chip are labeled as red arrows. SSX data collection were conducted as pump-probe system with specific time-delay between laser pulse and a train of 100 ps pink-beam x-ray pulses. To avoid the impact of zinc diffusion on data collection the step size of the SSX crystal holder was increased for the detection of longer time-points of moxalactam cleavage reaction. **d** Microscope photography of x-ray burn paper after test run of pump-probe SSX system.

electron density for the protein, bound metal ions and a ligand molecule. Throughout all time points the protein shows rather remarkably small conformational changes, including majority of residues in and near the active site.

### Room temperature crystal structure of *S. maltophilia* L1 MBL

The structure of L1 from *S. maltophilia* adopts a typical MBL fold with two β sheets decorated with exposed α-helices and loops (Fig. 2a). The active site of the L1 is located on the edges of β sheets and is surrounded by several loops. It possesses the most common active site motif for MBLs and comprises a sequence motif H116-X-H118-X-D120-H121[8]. The zinc ions are coordinated by H116, H118 and H196 (termed Zn1) and D120, H121 and H263 (termed Zn2). At physiological pH, the active site is positively charged due to a presence of zinc ions and partly protonated histidines (Fig. 2a). Due to similarity of MBL subclasses in overall protein fold and arrangement of key residues in active sites observed here mechanistic details of L1 reaction could be generalized not only for B3 (FEZ-1) representatives but also B1 (NDM-1), B2 (SFH-I) subclasses (Fig. 2b, c).

The "pink" reference structure of L1 MBL in the complex with the hydrolyzed moxalactam determined using SSX technique is virtually identical to the structure previously solved at high resolution at 100 K using single crystal methodology (Fig. 2). Root mean square deviation of superimposed Cα positions between L1 structures determined using "pink" beam SSX at 295 K (PDB entry 7L91), and the structure determined at 100 K (PDB entry 6UAC) is 0.32 Å (Supplementary Fig. 3d). The L1 both in solution, as determined by size exclusion chromatography, and in crystals is a tetramer (Supplementary Fig. 2a). In the crystal form used in all SSX experiments, four subunits of L1 tetramer are related by crystallographic symmetry.

### Snapshots of L1 biding events and cleavage of β-lactam ring

We recorded the time course of the $Zn^{2+}$ ions and moxalactam binding, chemical transformation, and conformational changes for L1 MBL (Fig. 3). Moxalactam (Latamoxef) is a broad-spectrum third generation cephalosporin of the oxacephem family, except that the sulfur atom of the cephalosporin dihydrothiazine ring is replaced with an oxygen atom to generate dihydro-1,3-oxazine. Moxalactam core has double ring architecture with a 4 membered β-lactam ring fused with a 6 membered oxazine ring. The double bond in moxalactam dihydrooxazine is formed between C-2 and C-3 flattening oxazine ring conformation (Fig. 3a). In our structures, the moxalactam core (β-lactam and oxazine rings (oxa-β-lactam)) is well ordered and shows good electron density for all TR-SSX structures. The moxalactam core is modified using three moieties with significant functions in antibiotic biochemistry and cell activity: the methyltetrazolethio constituent maximizes in vitro activity, the 7-alpha-methoxy group confers β-lactamase activity and the p-hydroxyphenylmalonyl moiety improves pharmacokinetics and half-life without high serum binding[47]. In our structures, these substituents are less ordered, particularly, distal parts of these moieties show relatively poor electron density, indicating flexibility that is likely due to fewer interactions with protein in the open to the solvent enzyme cleft. This is consistent with previous observations that the enzyme recognizes a substrate through binding an intact β-lactam moiety[23]. This also explains enzyme promiscuity, as it can accept multiple substrates and cleave them efficiently. The open active site with flexible loops allows for fast association, cleavage, and dissociation events.

During the L1 MBL catalysis, opening of the β-lactam ring of moxalactam occurs between C-8 and N-5 atoms (Fig. 3a). Computational modeling of the cleavage reaction progression in solution shows that β-lactam ring hydrolysis is completed within 2000 ms[24]. Using this information, we designed TR-SSX experiment that covers entire chemical transformation with snapshots that are approximately regularly spaced on logarithmic time scale (Fig. 3a, b). We collected TR-SSX 295 K data at 20, 40, 60, 80, 100, 150, 300, 500, 2000, and 4000 ms. In addition to these ten time-points, we solved several SSX structures at 295 K including apo enzyme, one with single zinc ion and one with two zinc ions as well as completely hydrolyzed moxalactam ("pink" reference). These structures serve as references for TR-SSX experiments

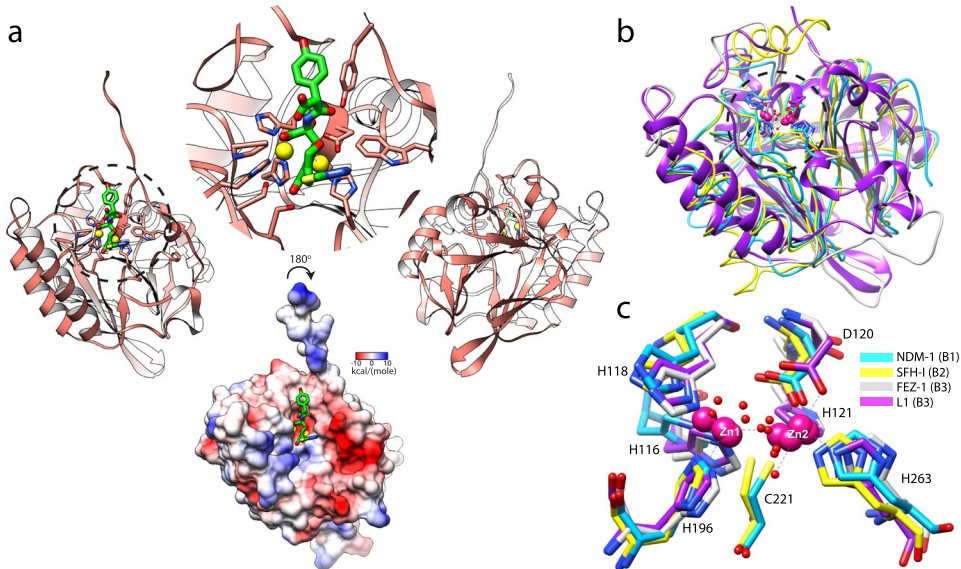

**Fig. 2 | Overview of crystal structure of L1 from *S. maltophilia* in a complex with hydrolyzed moxalactam. a** Structure of L1 is shown as cartoon (salmon), active site contains moxalactam (green sticks), and two zinc ions are showed in yellow. Bottom surface depicts the Coulombic electrostatic potential plotted on the surface of the L1 in complex with moxalactam at 295 K (electrostatic potential calculated using default settings in UCSF Chimera). **b** Cartoon representation of L1 MBL superposed with crystal structures of B1, B2, and B3 subclasses of MBLs depicted as licorice models. Crystal structure of L1 MBL determined at room temp using SSX (PDB entry 7L52 - magenta) was superimposed with structures of MBL representatives: B1 subclass NDM-1 from *K. pneumoniae* (PDB entry 6TWT - cyan), B2 subclass mono-zinc MBL SFH-I from *Serratia fonticola* (PDB entry 3SD9 - yellow), B3 subclass FEZ-1 MBL from *Legionella gormanii* (PDB entry 5WCK - gray), two zinc ions are showed in pink. **c** Comparison of key residues in the active site of MBL subclasses. Zinc atoms showed as pink spheres, water molecules are depicted as red spheres with a small radius.

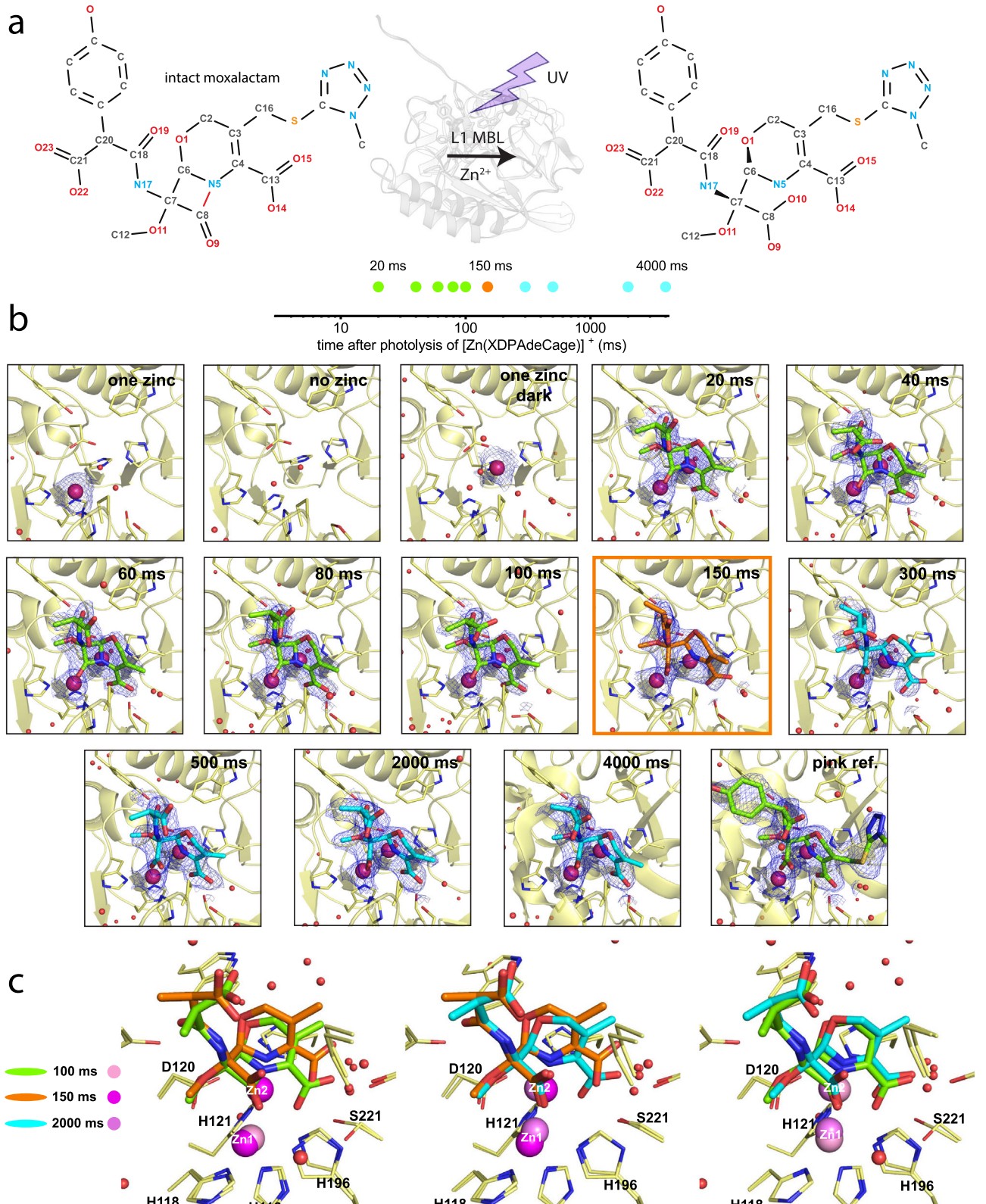

**Fig. 3 | TR-SSX structures of moxalactam bound to the active site of L1 MBL.**
**a** Schematics of moxalactam hydrolysis and atom numbering scheme. **b** Room temperature SSX and TR-SSX structures of L1 active site representing different states (with one zinc, with no metal (both EDTA treated), with one zinc ion bound "dark" structure, time points from 20 to 4000 ms and "pink" reference with two zinc ions and fully hydrolyzed moxalactam). His263 coordinating Zn2 is not displayed for clear view of moxalactam. The 2Fo-Fc map (blue mesh) contoured at 1.0

σ level (carved at 1.4 Å) around moxalactam and zinc ions in catalytic site of L1 MBL. **c** Substrate, intermediate and product of moxalactam hydrolysis by L1 are shown at three time points. Representation of three key steps captured during reaction (100 ms bound uncleaved moxalactam – substrate (green), 150 ms hydrolyzed moxalactam intermediate (orange), 2000 ms structure with hydrolyzed moxalactam after reaction – product (aqua)). Legend on left site of image illustrates shades of pink spheres that visualize zinc atoms.

(Fig. 3b, Supplementary Fig. 4a). The "dark" structure was obtained with crystals in the presence of [Zn(XDPAdeCage)]+ and moxalactam, but no laser exposure. In this structure we observe partly occupied zinc ion in Zn2 site, suggesting that the metal cage was somewhat leaky, however this structure does not contain the electron density for moxalactam (Fig. 3b). It was reported for L1 to bind one zinc ion with higher affinity and this site was sensitive to presence of substrate[48]. Several high-resolution structures determined previously at cryo-conditions (100 K) assist us in the interpretation of the TR-SSX data[9].

## The TR-SSX reveals changes in interatomic distances during reaction

The TR-SSX experiments showed the time course of reaction starting with empty L1 active site, binding two $Zn^{2+}$ ions, followed by moxalactam binding, repositioning substrate in the active site, cleaving moxalactam and conformational adjustments of the reaction product (Figs. 3 and 4, Supplementary Figs. 4c and 6). All these variations are revealed by changes in the distances between protein, zinc and ligand atoms (Supplementary Tables 3 and 4). In the first data point collected at 20 ms, two bound $Zn^{2+}$ ions are observed and

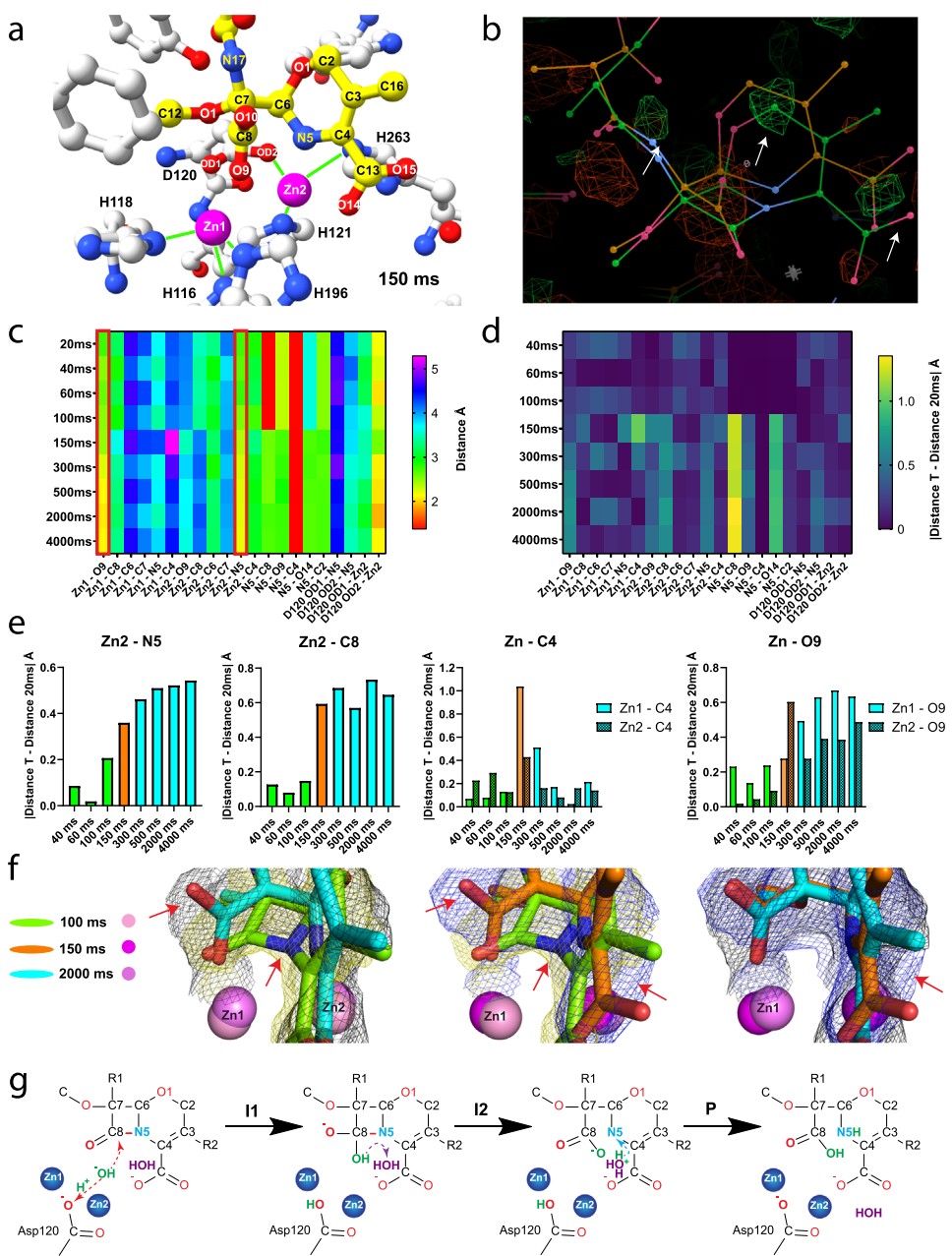

**Fig. 4 | TR-SSX crystal structures of moxalactam of the active site of L1 MBL.** **a** L1 active site structure at 150 ms with hydrolyzed moxalactam (in yellow-red-blue), zinc (magenta) and protein residues (in silver-blue-red). **b** Difference electron density map calculated as $F_{obs}$(150 ms)-$F_{obs}$(100 ms). The differential map was calculated with Phenix isomorphous difference maps module. Negative electron density showed as red mesh, positive as green mesh. Depicted in the Coot electron density map are countered with a 1.9 σ level. Orange lines illustrate cleaved moxalactam at 150 ms, green lines represent uncut moxalactam at 100 ms. **c** Distances between atoms key in description of L1 induced β-lactam cleavage reaction

(rainbow). **d** Distances in TR-SSX structures of L1 MBL relative to atom positions in 20 ms crystal structure of L1 MBL (illustrated as a heat-map). **e** TR changes in distances between zincs and moxalactam atoms modeled in L1 active site, source data is available as Source data file. **f** Evolution of electron density during cleavage of moxalactam by L1 MBL. The 2Fo-Fc map contoured at 1.0 σ level (carved at 1.4 Å) around moxalactam; maps were depicted as olive mesh for 100 ms, blue mesh for 150 ms, and black mesh for 2000 ms. **g** Catalytic mechanism proposed for L1 MBL based on TR-SSX experiments.

partial electron density for a moxalactam molecule is also detected (Supplementary Movie 2). Zinc ions and antibiotic molecule displace several water molecules which were present in the "dark" structure. The electron density for an unhydrolyzed moxalactam increases and remains visible in the next four time points (40, 60, 80, and 100 ms) and the ligand molecule adjust its conformation, as shown by shifting of atom positions (Fig. 4c, d, Supplementary Fig. 4c, Supplementary Tables 3 and 4). The occupancy of Zn2 site is higher than the Zn1 and is typically equal or higher than substrate. The important role of Zn2 in substrate binding was observed by Rasia and Vila[12]. The Zn – Zn distance initially at 3.2 Å (40–100 ms), increases to 3.4 Å near cleavage time (150 ms) and becomes the shortest at steady state (3.0 Å, 2000 ms). At the 150 ms time point, after cage photolysis, we detected an opened β-lactam ring of moxalactam, revealing the structure of a potential intermediate. This intermediate has been originally proposed by Benkovic[18] and Crowder[49] for nitrocefin, and later by Vila for carbapenems[27], who later elaborated that the two main steps of reaction (C–N bond cleavage and the protonation of the nitrogen atom) were synchronous in his review[20]. Moreover, the part of the ligand was significantly displaced at the active site (Fig. 3c). The electron density for the hydrolyzed moxalactam continued to be visible in the 300, 500, 2000, and 4000 ms time points (Supplementary Movie 2). The chemical reaction is completed within 50 ms (100 ms–150 ms) in majority of molecules in the crystal (no mixture of states can be refined at 100 and 150 ms). The two steps of reaction seem synchronous as observed by Vila et al.[50]. The hydrolyzed moxalactam molecule undergoes further conformational tuning until reaching steady state at 2000 ms and it adopts a conformation similar to that of the unhydrolyzed ligand at the 100 ms time point (see the third picture of Fig. 3c). Interestingly, the 2000 ms time point has the lowest B-factor values for ligand 28.8 Å$^2$, protein 25.10 Å$^2$ and Zn2 and Zn1 ions (14.20, 27.96 Å$^2$, respectively). Values of B-factors at 2000 ms indicate that the structure of L1 after reaction is relaxed in the most stable conformation (Supplementary Fig. 5, Supplementary Tables 5 and 6).

During the time leading to the bond breakage (between 20 and 100 ms) there is no significant distance changes between ligand and protein as indicated in Supplementary Table 4 and depicted on Fig. 4c, and Supplementary Fig. 4d. This is consistent with the idea that the substrate molecule fluctuates slightly prior to the catalytic event. This also includes oxazine as the same ring configuration and bond distances are always maintained in the structures in all time points. After the bond scission occurs between C-8 and N-5 atoms there are some significant distance changes, and the product settles down by binding to two zinc ions and stays bound till 4000 ms time point. Unexpectedly during catalysis, the oxazine ring rotates counterclockwise (~30 degrees) along the C-6–C-7 bond as scissile bond gets broken between 100 and 150 ms (Supplementary Fig. 4d, e). The oxazine ring is also flatter at 150 ms than at other time points. Then the ring rotates back clockwise from 500 to 4000 ms and returns to the configuration similar to that of the intact substrate. In the process, atoms N-5 and O-14 (as the C-13 carboxyl turns slightly) approach Zn2 (reducing distance from 2.4 to 2.2 Å and 2.9 to 2.3 Å, respectively) (Supplementary Fig. 4e). The product is then stabilized and stays bound to both zinc ions. Right after the scission event when a hydroxyl molecule attacks carbonyl C-8, newly formed C-8 carboxyl moiety is well visible in electron density (Fig. 4f). The O-10 atom most likely comes from the attacking hydroxyl molecule (not from the carbonyl O-9) as it is located closer to the Zn1. Interestingly all possible residues involved in catalysis (Zn2, His and Asp) are located near Zn1. Because of the 2.20 Å resolution of the structures it is not possible to address possible molecules tautomeric states during the catalysis.

This is in agreement with previously published EXAFS and XANES data where the shortest Zn-Zn distance is observed in resting state and the longest near the catalytic event as coordination sphere of zinc ions changes[11,25]. It has been reported that the distances vary with pH and are shorter at high pH[23]. Our TR-SSX experiments were performed at pH 8.0 and Zn-Zn distances are consistent with this observation.

## L1 MBL conformation during reaction

Strikingly, in contrast to the variations in ligand atoms, the protein shows very limited shifts in positions of amino acid side chains, metal center, and solvent structure in the active site during catalytic process (Fig. 4). We observe only small adjustments in the conformation of the residues that directly coordinate di-metal scaffold (H116, H118, D120, H121, H196, and H263) (Fig. 3c). However, we detect different rotamers of some site-chains in the active site, namely S221, I162 and some small shifts of the loop formed by the L1 residues between S221 and G232 (Fig. 3b). S221 shows three distinct conformations. Most of the time S221 assumes trans conformer with exception of two time points, one just prior cleavage (100 ms) and the second after cleavage (300 ms). This residue forms direct hydrogen bond with the C-4 carboxylate of the moxalactam during the reaction and seems to tract its movement in different stages of catalytic process and conformational adjustments (Fig. 4). We believe it is significant because these time points are just before and after β-lactam ring cleavage, respectively and are associated with conformational adjustments of the substrate and the product in the active site. Other residues near the active site showing small conformational changes include I162, which displays three different rotamers and S223. These side chains most likely respond to local volume changes in the active site during reaction. There are several charged residues nearby showing multiple conformations in TR-SSX structures (R30, K300, and D35) but they do not interact with the ligand, nevertheless they may alter electrostatic near the active site. Y32 also shows small conformational adjustments associated with ligand shifts as it interacts with the carbonyl at C-18 of moxalactam at the fringe of the active site.

Water molecules are essential for catalysis. Ordered water molecules were identified and placed in the vicinity of the active site. They form hydrogen bonds with the ligand (C-4 carboxylate) and the protein (S221) at 150 and 300 ms time points. However, none of these waters interact directly with metal center. Two water molecules (W423 and W470) are within 7 Å radius from bi-zinc center just prior β-lactam cleavage. This pool of solvent shows somewhat altered positions in different structures. We believe these waters can rapidly approach the zinc scaffold and be activated for nucleophilic attack on C-8 (Fig. 4). Interestingly, W423 in gone after reaction at 150 ms. Our previous MD simulations for NDM-1 indicated low energetic barrier for such activation[23]. In L1's case after the UV laser irradiation, zinc ions are released from photocages and bind to active site, water molecules and some side chains in active site can move, perhaps on time scales much faster than our TR-SSX experiment.

L1 crystallizes in the presence of zinc. Crystals of L1 can be treated with EDTA for different periods of time to remove one or both zinc ions. Shorter time treatment resulted in the single zinc ion bound (Zn2). Longer time treatment removed zinc ions completely resulting in the "empty" active site containing only ordered water molecules (Supplementary Fig. 4a). The reaction progress recorded in the TR-SSX experiment shows small variations in occupancy and B-factors of zinc ions (Supplementary Table 5). The "dark" structure has been refined with one zinc ion (Zn2) that possesses relatively high B-factor 62.03 Å$^2$.

Our structures significantly contribute to understanding the mechanism of the L1 MBL activity. It appears that the enzyme catalytic site with Zn$^{2+}$ bound is in an "active" state prior to ligand binding and is presenting a metal scaffold for a substrate. It is the ligand that must adjust its conformation for a catalytic event to occur (Supplementary Fig. 6). Some distances between atoms change very little during the entire process (Zn2 – C-6, Zn1 – C-8, N-5 – C-2) (Supplementary Tables 3 and 4). However, we observe some very dramatic and distinct changes in the distances between protein and ligand atoms during

catalysis. The C-4 carboxylate, C-8 carbonyl and N-5 of β-lactam are directly binding zinc ions, with one oxygen atom of the C-4 carboxylate interacting with both zincs (Zn1 – C-4, Zn1 – O-9, Zn2 – C-8, Zn2 – O-9, N-5 – C-8, N-5 – O-14). This carboxylate also hydrogen bonds with H196, S221 and S223. After reaction some of these distances return to original state (Zn1 – C-4, Zn1 – N-5), but some are now in a new state (Zn1 – O-9, Zn2 – C-8, N-5 – C-8) consistent with completion of chemical transformation (Fig. 4c, d, e, Supplementary Tables 3 and 4). Certain shifts may reflect formation of intermediates.

The L1 uses two-zinc scaffold that is central to MBL's activity, contributing to active site stability, substrate binding, and catalysis. The β-lactam substrates that are hydrolyzed by L1 all possess carboxyl and carbonyl groups that are spaced by their similar, roughly planar core geometries. Oxygen atoms from these groups coordinate with the zinc ions to form the principal substrate-recognition unit of the enzyme. The change in a distance between zinc ions can be accommodated by rotation of carboxylate. Therefore, only when two zinc ions are present a substrate can bind, but reaction does not proceed immediately. Zn2 seems crucial for strong β-lactam binding as it positions the C-8 atom for nucleophilic attack. For L1 these binding events and early adjustments occur on the time scale of 100 ms for majority of L1 molecules in the crystals. During this time, conformation of substrate is slightly adjusted and there is small zinc ions movement and solvent positions shifting. A water molecule must be activated before nucleophilic attack on C-8. There is no electron density found for the water bridging two zinc ions in any of our TR-SSX time points structures (Fig. 3b, Supplementary Movie 2). The only structure that shows bridging water molecule is the "pink" reference structure (Fig. 3b, Supplementary Fig. 4b), corresponding to enzyme/hydrolyzed moxalactam complex after the reaction is completed. In the unhydrolyzed moxalactam state the oxygen of C-4 carboxylate bridges both zinc ions raising possibility of substrate assisted catalysis. There is no trace of electron density corresponding to attacking water molecule near zinc scaffold, yet the reaction occurred, the β-lactam ring was hydrolyzed and a new carboxylate forms on C-8 with oxygen atom from attacking water present. Interestingly, as the protein surface is predominantly hydrophobic, the di-metal scaffold provides enticing pull for water molecules. A water molecule can be attracted by di-zinc center, and it becomes "attacking" hydroxide and it rapidly reacts with C-8 that is precisely positioned for reaction. The di-nuclear zinc catalytic core is coordinated by six side chains (five His and an Asp residues) that form 2 catalytic triads. The presence of the zinc ions provides a pathway for shuttling protons among the water molecules that would not be present in bulk water, explaining the relative stability of the substrates in solution. The bulk water molecule transiently becomes oriented and can serve as the general base. D120 binds to both zinc ions and seems to be in position to strip proton from incoming water molecule. However, nucleophile activation by metal ions does not require a general base to deprotonate the water molecule. Therefore, the role of D120 together with conserved residue H118 is to position Zn2 for substrate binding and catalysis. The incoming water (for example W423) is activated and resulting hydroxide attacks C-8 and forms intermediate anion(s) (Fig. 4g). Ring cleavage is completed through a proton translocation process that possibly utilizes another bulk water molecule as it was suggested previously[23]. The protonation of the N-5 nitrogen atom may be the limiting step. In 100 ms structure there are two water molecules that are closest to zinc ions (W423 and W470), and they may be good candidates for the reaction. These events must occur on a short time scale and were not captured in our TR-SSX experiment. The proposed mechanism is in agreement with EXAFS data suggesting a multistep reaction that seems synchronous[11]. Once the scissile carbon-nitrogen bond is cleaved the reaction is over, C-4 carboxylate moves away from the Zn2 site and it interacts only with Zn1 (Figs. 3, 4). Newly formed C-8 carboxylate binds both zinc ions. Hydrolyzed moxalactam continues to adjust its

conformations (for example Zn1 – O-9, Zn2 – N-5) until it reaches steady state at 2000 ms.

## Conclusions from SSX and TR-SSX experiments

The role of metal ions in binuclear MBLs has been studied extensively[9–11]. The proposed catalytic mechanism suggests that the metal scaffold plays a key role in substrate recognition and binding, in activating the water/hydroxyl molecule for the nucleophilic attack and electrostatic stabilization of the negative charge of proposed anionic intermediates.

Our TR-SSX experiments using β-lactam antibiotic and L1 MBL enables us to visualize progression of binding metal ions and substrate to the active site of enzyme. The metal scaffold is essential for recognition β-lactam portion. Our structures show that interaction of other moieties of the ligand with protein are secondary and explain enzyme specificity for β-lactams and their ligand promiscuity. However, upon substrate binding the chemical transformation does not occur instantly but it requires ligand conformation adjustments in the active site. There is also no bridging water observed bound to zinc ions. Once the ligand achieves suitable conformation in the active site the enzyme is ready to attract and activate the water molecule to hydroxyl that can rapidly attack the C-8 atom. The proposed anionic intermediates formed in the first step of the chemical reaction are likely stabilized by the positive charge of the zinc scaffold. The metal scaffold, C-4, D120 carboxylate, water molecule and nearby amino acid sidechains (H121, H181, S221, H263) provide environment to shuffle proton needed to complete chemical transformation. During chemical transformation the distances between atoms of substrate and protein, including zinc ions change significantly. The newly formed carboxylate on C-8, containing oxygen from attacking hydroxide is stabilized by the interaction with Zn2. Unexpectedly during catalysis, the oxazine ring on moxalactam rotates back and forth revealing new conformational state. Our data is consistent with the mechanism proposed by Lisa et al.[11] for binuclear MBLs with further refinement of this mechanism as all chemical transformations for L1 occur in less than 50 ms and all intermediates are short-lived and two steps of reaction appear synchronous. It was suggested that the product of β-lactam cleavage by MBLs may undergo tautomerization[51], however, resolution of our data does not allow us to resolve these states. Much finer time slicing experiment and higher resolution data are needed.

The L1 active site (B3 subclass) is very similar to the NDM-1 (B1 subclass) and possesses structural arrangement most common in MBLs. The importance of Zn2 site underscores its key role in subclass B2 enzymes. The mechanistic details of L1 captured during the reaction should be generalized to majority of B1 and B3 MBLs. These observations can aid development of more effective inhibitors against emerging pathogens with antibiotic resistance based on activity of MBLs. Interestingly, recent report that 2-mercaptomethyl thiazolidines (MMTZ) can inhibit all three subclasses on MBLs suggest that these enzymes share common and conserved ligand binding mode[52]. Structures reveal that inhibition involves direct interaction of the MMTZ thiol with the mono- or dizinc centers of B1, B2 and B3 enzymes. We expect that the ligand conformations that are the closest to the catalytically compatible state would correspond to the best inhibitor conformation that would fit into active site. Furthermore, the discussed methodology of the TR-SSX with reaction triggered by photolysis of [Zn(XDPAdeCage)]+ could be applied to investigate other metalloproteins and can become common technique for study of dynamics of other metal dependent enzymatic systems.

The detailed understanding of the reaction mechanism is essential to design new inhibitors with different stereochemistry. The further development of SSX and implementation of TR-SSX is an approach that could reveal time-dependent molecular details of the enzymatic activity of MBLs and describe the complete mechanism of nucleophilic water attack that cleaves the β-lactam ring. Investigating

new approaches to study MBLs could lead to discovery of effective inhibitors against *Enterobacteriaceae* and other pathogens that carry genes of MBLs. The TR structural studies could reveal states of enzyme that are non-accessible using other static techniques and can support computational modelling. Therefore, drug discovery based on the details captured by high resolution crystal structures could contribute key information to counteract antimicrobial resistance and provide tools in the next pandemy[53].

## Methods

### SSX data collection and processing at 19-ID

For this study, we determined two structures using SSX: one with single zinc ion at Zn1 position, and the second structure of L1 depleted of zinc ions in the active site. For these structures, crystals were treated with EDTA for various periods of time. Recently we also determined using SSX, high-resolution room temperature structure of L1 MBL with two zinc ions bound (PDB entry 7L52[36]). To collect SSX data we used fixed-target serial crystallography platform developed at the Structural Biology Center (SBC) 19-ID beamline at the APS[45,46]. The diffraction data were recorded at 295 K using the *ALEX* SSX platform on the PILATUS3 X 6 M detector placed at the 350 mm distance from sample. In brief, data were collected with SBCcollect software with a fixed-target chip step size of $50 \times 50\,\mu m$ ($75 \times 75\,\mu m$), x-ray beam sizes of $50 \times 50\,\mu m$ ($75 \times 75\,\mu m$), and sample exposure on x-ray pulse of 40 ms (60 ms) (values in brackets for structure determination of L1 with zinc ions). Diffraction data for determination of L1 without zinc, and with zinc ions were collected from single chip for each data set. Overall, 38,500 and 18,000 images were acquired, respectively. Using *DIALS* and *PRIME* software[54] we indexed, and integrated 5191 (3374) diffraction images per data set. The average crystal hit rate for the two SSX data sets was 16.2%. After the first run of *PRIME* data were subjected to outlier removal, which rejects images with resolution lower than 2.0 (2.2) Å. After outlier rejection, a final count of 4550 and 2947 diffraction images were merged for L1 crystal structure determination without zinc, and with zinc ions, respectively.

### Time-resolved SSX data collection at BioCARS 14-ID

The polychromatic TR-SSX data were collected at the BioCARS 14-ID-B beamline at the APS[55]. We used the same fixed-target system as described for 19-ID data collection, with crystal slurry loaded on the *ALEX* holder. The polychromatic x-rays (1.02–1.18 Å wavelength range, with the bandpass 5% or 600 eV (FWHM)) were focused using Kirkpatrick–Baez mirrors[55] to beam size 30 μm × 30 μm. The average crystals size of L1 MBL used for SSX data collection was $20 \times 20 \times 100\,\mu m$. Diffraction data were collected during two APS runs, exposing each crystal on the *ALEX* chip to either 24 or 48 consecutive x-ray pulses (pulse train durations of 3.7 μs or 7.4 μs, respectively) (Fig. 1, Supplementary Table 1). For calculation of the x-ray dose delivered to crystals we used RADDOSE-3D v4.0.1011[56]. The pulse trains were isolated by an ultra-fast Jülich chopper. A millisecond shutter was used to select a single opening of the chopper[55]. Diffraction images were recorded on RAYONIX MX340-HS detector. The detailed information about polychromatic diffraction data collection parameters is listed in Supplementary Table 1.

In the TR-SSX experiments, zinc ions were released from the [Zn(XDPAdeCage)]+ at the beginning of each time point by a pulse of UV-light at 347 nm resulting in the photodecarboxylation of the cage. We used nanosecond pulses from an OPOTEK Opolette 355 II HE laser, focused to $100 \times 80\,\mu m$ spot. To analyze a specific time-point of the reaction we used pump-probe technique. The chip moved to a specific position, then the sample was irradiated using laser pulse (7 ns), after a specific time delay (20–4000 ms) the x-ray shatter was open and diffraction image was collected. The chip moves through the x-ray beam in a meander pattern covering the area of the chip (Fig. 1). To avoid

impact of de-caged zinc ions diffusing during the time delay through crystals on the *ALEX* chip, we used several step sizes during chip scan. Depending on the time-point, the fixed-target chip moved horizontally from 200 to 450 μm, and vertically from 250 to 450 μm, respectively for 20 ms and 4000 ms delay (Supplementary Table 1, Fig. 1). Depending on the time-points, the total time of data collection from single chip varies from ~20 to 70 min, respectively for collecting 20 ms, and 4000 ms data.

### Polychromatic x-ray diffraction data analysis

For analysis of "pink" beam TR-SSX images we used the pyPrecognition Python script that finds diffraction patterns containing a specified number of spots above the background threshold level (hits). For spot finding we used parameters from 30/25 to 50/60 depending on the data set (Supplementary Table 1), the first numeral is the number of spots above the minimum intensity and second numeral is the minimum intensity above the background of a spot to be considered a diffraction spot. After initial selection of hits, Laue serial crystallography data were indexed in the Precognition software (Renz Research, Inc.). The indexing of Laue data was based on space group and the cell unit parameter obtained for the 295 K monochromatic SSX L1 MBL structure (PDB entry 7L52). Subsequently, indexed images were checked both automatically and visually to eliminate multiple diffraction patterns. Some images automatically reported as indexed were mis-indexed and rejected from analysis after inspection. These procedures reduced the initial number of indexed images to approximately one third of initial images (Supplementary Table 1). Selected images were further processed, cell parameters were refined, and data were integrated to 2.2–2.4 Å resolution depending on the time-point. Data were scaled and merged in Epinorm software (Renz Research, Inc) with the threshold level of $I/\sigma(I) > 3.0$.

### SSX and "pink" beam TR-SSX structure determination

Structures of L1 MBL were determined using molecular replacement (MR) with MOLREP[57] implemented in the CCP4 software package[58]. To determine the structure of L1 using monochromatic SSX diffraction data we used the 1.60 Å resolution crystal structure of L1 in the complex with moxalactam as a search model for MR. This structure was previously determined at 19-ID using single crystal at 100 K (PDB entry 6UAC)[9].

For TR-SSX data we used an HKL output file containing amplitudes (Fs and σFs) after data scaling in Precognition/Epinorm to convert to MTZ file with Fs, σFs and $R_{free}$ flags in CCP4 software package[58]. After the MR step (using MOLREP) structures of L1 MBL were initially refined in REFMAC[59]. Finally, all structures obtained using monochromatic SSX, and polychromatic TR-SSX were refined with Phenix.refine by both rigid-body and regular restrains refinement[60]. The procedure of refinement in each case started with simulated annealing for the first three runs of refinement. Initially, the model without a ligand molecule was refined, secondly zinc atoms were placed in the active site, and before the third refinement the ligand molecule was added to the refined model. The ligand molecule was placed in the electron density calculated from the coordinates of the L1 MBL structure in complex with moxalactam determined for the "pink" beam reference structure (PDB entry 7L91) by superposition of Cα atoms of the proteins. L1 MBL structures were iteratively refined in Phenix followed by model adjustments in Coot[61]. Water molecules were generated using the ARP/wARP[62] automated procedure, and inspected manually in Coot. The stereochemistry of L1 MBL structures were validated using Ramachandran plot and MolProbity[63]. Detailed statistics of data refinement are presented in Supplementary Table 2. For presentation of molecular models we used Pymol and UCSF Chimera[64]. The heat-map illustrating changes of distances between key atoms over time was plotted using GraphPad prism version 8.

## Expression and purification of L1 MBL

The gene cloning was performed according to Kim et al.[9]. The L1 gene (residue 20–290) from *S. maltophilia K279a* was amplified from the genomic DNA with KOD Hot Start DNA polymerase. For the construct 5′ TACTTCCAATCCAATGCCAGCGCCGCCGAGGCAC forward and 5′ TTATCCACTTCCAATGTTAGCGGGTCCCGGCCGTTT reverse primers were used. The PCR products were purified and cloned into the pMCSG53 according to the ligation-independent procedure[9] and transformed into the *E. coli* BL21 DE3 (Gold) strain (Stratagene). A single colony was picked, grown and induced with isopropyl-β-D-thiogalactoside (IPTG). The cell lysate was analyzed for presence of the protein with the correct molecular weight. The transformed cells were grown at 37 °C until reaching optical density at 600 nm equal 1.0 OD. Subsequently, cell culture was cooled down to 18 °C, protein expression was induced with 0.2 mM Isopropyl β-D-1-thiogalactopyranoside and cells were grown overnight. Bacterial cells were harvested by centrifugation at ×7000 *g* (Sorval evolution RC centrifuge, Thermo-Scientific) and cell pellets were resuspended in a 35 ml lysis buffer (500 mM NaCl, 5% (v/v) glycerol, 50 mM HEPES pH 8.0, 20 mM imidazole and 10 mM β-mercaptoethanol) with one tablet of protease inhibitor cocktail (Complete Ultra-EDTA-free, Sigma) per liter culture and treated with lysozyme (1 mg/ml) and lysed by sonication and centrifuged at ×29,500 *g*. The supernatant was applied to Immobilized Metal Affinity Chromatography (IMAC) 5 mL HisTrap™ Fast Flow column (cytiva) and washed with lysis buffer. The protein with His$_6$ affinity tag was eluted from the column using the lysis buffer supplemented in 500 mM imidazole. We used the Midwest Center for Structural Genomics (MCSG) protocol for automated purification of proteins on ÄKTAxpress systems (cytiva)[65]. Supernatant obtained from 12 L cell culture was purified on 4 HisTrap™ Fast Flow columns to avoid column overloading. We used Tobacco Etch Virus (TEV) protease at 1/40 molar access to remove His$_6$ tag attached to N-terminus of recombinant L1 MBL. Incubation with protease was done overnight at 8 °C. A second IMAC purification was used for removal of cleaved N terminal His$_6$ tag, contaminants that binds to IMAC column, and TEV protease that also possess uncleavable His$_8$ tag. In the case of the second IMAC we collected flow through sample which contains pure L1 MBL without a His$_6$ tag. The sample was dialyzed against a protein buffer A (15 mM HEPES pH 7.0, 100 mM KCl, and 2 mM TCEP). Purified L1 MBL was concentrated to 95.8 mg/mL using centrifugal concentrator Amicon® Ultra-15 (Merck) with cut-off 10 kDa, and cryo-cooled using 35–40 μL aliquots dropped into liquid nitrogen. The total amount of L1 MBL obtained from 12 L culture was 1100 mg, sufficient for subsequent optimization of batch crystallizations, and preparing multiple crystallization samples for TR-SSX experiments.

## Batch crystallization of L1 MBL

The crystals of L1 MBL were produced from a single batch of purified protein using the protocol listed below. We used the same batch crystallization setup for the SSX experiment at 19-ID beamline, and for the TR-SSX experiment at 14-ID-B beamline at the APS. The batch crystallization was controlled by seeding. The L1 MBL aliquots stored at −80 °C were rapidly thawed to RT and the sample was diluted to 47.9 mg/mL in buffer A supplemented with 5 mM ZnCl$_2$. L1 MBL sample was mixed with crystallization solution containing 0.15 M sodium malonate pH 8.0 (Hampton Research), 20% (w/v) PEG3350 (Hampton Research). Crystal nucleation was initiated by adding freshly prepared seeds. Seeds were generated from a single crystal with an approximate dimension of 30 × 30 × 300 μm transferred to 200 μL of crystallization buffer using PTFE Seed Bead kit (Hampton Research). The tube was vortexed at 3000 rpm for 2 min, subsequently crystal seeds were added to batch crystallization in a volume ratio 1:100. Each batch crystallizations sample contained 400 μL solution in 1500 μL polypropylene tubes and stored in the horizontal position in a 289 K crystallization chamber. First crystals appeared after 3 days, reaching the final dimensions of 20 × 20 × 100 μm 2 weeks later. These crystals were suitable for SSX experiments.

## Crystals handling and preparation [Zn(XDPAdeCage)]$^+$ for TR-SSX

Batch crystallizations were started 3 weeks before planned beam-time at the light source. Crystallization of L1 MBL requires presence of zinc ions in a buffer to grow high symmetry space group P6$_4$22 crystals. However, for the TR-SSX experiment, metal had to be removed from crystals. This was accomplished by washing crystals in EDTA solution 3 days prior to the TR-SSX experiment. Crystals of L1 MBL were collected in Eppendorf tubes by centrifugation for 2 min at 100 RPM in RT, the crystallization solution was replaced with 600 μL of buffer B containing 0.06 M sodium malonate pH 8.0, 28% (w/v) PEG3350 supplemented with 1 mM EDTA. After 2 h we performed a second crystal wash using 1400 μL of the buffer B. Subsequently, after 24 h, a third crystal wash was done with 600 μL of buffer B containing 1.5 mM EDTA. The next day crystals were rinsed with 1000 μL of buffer B without EDTA.

For TR-SSX, the EDTA treated crystals of L1 MBL were additionally supplemented with [Zn(XDPAdeCage)]$^+$ developed by the Burdette laboratory[44]. The XDPAdeCage was at first solubilized in DMSO at 250 mM, subsequently ZnCl$_2$ was added to saturate 60% capacity of the XDPAdeCage binding sites. Samples were thoroughly mixed and incubated overnight. The [Zn(XDPAdeCage)]$^+$ was diluted five times in water. Subsequently, the sample was equilibrated with buffer B without EDTA in a 1:1 volume ratio. The solution containing 15 mM [Zn(XDPAdeCage)]$^+$ and 10 mM unsaturated XDPAdeCage was used for TR-SSX data collection. The UV induced release of zinc ion from the [Zn(XDPAdeCage)]$^+$ was tested in solution (Supplementary Fig. 1).

Prior to the TR-SSX experiment the EDTA treated crystals were collected in Eppendorf tubes by centrifugation for 2 min at 100 RPM in RT, excess buffer was removed. Crystals were resuspended in 45 μL of residual buffer, 12 μL of 40 mM moxalactam solution was added and then 16 μL of the mixture of [Zn(XDPAdeCage)]$^+$/XDPAdeCage prepared as described above was added. Final concentration of moxalactam was 6.58 mM and the concentration of [Zn(XDPAdeCage)]$^+$ was 3.29 mM. The resulting volumes were sufficient for preparing three chips for SSX experiments. To perform fixed-target SSX experiment we used the ALEX holder (patent application #16/903,601)[46]. We used a 60 μm grid made from nylon (NY6004700, Millipore) that hold 15 μL of crystals slurry sandwiched between two layers of 6 μm mylar. The *ALEX* holders were assembled between 30 min and 1 h prior to exposure for the TR-SSX experiment.

## Data availability

The structural data generated during the current study are available in the Protein Data Bank repository (https://www.rcsb.org/) under accession codes: 7L91, 7UHR, 7UHS, 7UHT 7UHH, 7UHI, 7UHJ, 7UHK, 7UHL, 7UHM, 7UHN, 7UHO, 7UHP, 7UHQ. A Source Data for all tables and figures are extracted from PDB deposits listed above this manuscript. Plasmid for expression L1 MBL (in Vector pMCSG53 containing the L1 MBL gene) is available upon request. All other data generated during the current study including the raw biophysical data are available upon request. Source data are provided with this paper.

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

## Acknowledgements

We sincerely thank the members of the SBC at Argonne National Laboratory for their help with data collection at the 19-ID beamline and Paula Bulaon for help with preparing paper. Funding for this project was provided in part by federal funds from the National Institute of Allergy and Infectious Diseases, National Institutes of Health, Department of Health and Human Services, under Contract No. HHSN272201700060C. The use of SBC beamlines at the Advanced Photon Source is supported by the U.S. Department of Energy (DOE) Office of Science and operated for the DOE Office of Science by Argonne National Laboratory under Contract No. DEAC02-06CH11357. Results shown in this report are derived from work performed at Structural Biology Center which is funded by the US Department of Energy, Office of Biological and Environmental Research under contract DE-AC02- 06CH11357. Use of BioCARS was supported by the National Institute of General Medical Sciences of the National Insti-tutes of Health under grant number P41 GM118217. The content is solely the responsibility of the authors and does not necessarily represent the official views of the National Institutes of Health.

## Author contributions

M.W., D.A.S., Y.K., G.B., R.W.H., V.S., and A.J. designed the research. M.W., N.M., and M.E. purified L1 and crystallized protein. A.S., and S.C.B. synthetized the XDPAdeCage. D.A.S., A.L., R.W.H., K.L., and M.W. per-formed serial crystallography data collection. M.W., D.A.S., Y.K., V.S., and A.J. analyzed serial crystallography data. M.W., Y.K., A.L., and A.J. wrote the paper.

## Competing interests

The authors declare no competing interests.
