## [Peer Review File · Nature Communications]

REVIEWER COMMENTS

Reviewer #1 (Remarks to the Author):

Metallo- β -lactamases (MBLs) play crucial roles in conferring antibiotic resistance to pathogenic bacteria. Understanding the molecular mechanism of how these enzymes hydrolyze existing β -lactam antibiotics is of critical importance to drug discovery effort to develop novel anti-bacterial therapeutics.

The authors used time-resolved x-ray crystallography to track the hydrolysis of moxalactam in the active site of L1, an emerging MBL from the opportunistic gram-negative basillus pathogen *S. maltophilia*. Their time resolved synchrotron light source (TR-SSX) system is state-of-the-art, capable of tracking the catalytic process over the duration of 5-4000 millisecond. Altogether, ten structures representing the L1 enzyme at different time points of hydrolyzing moxalactam were obtained. These snapshots capture distinct steps of the catalytic process, including binding of the zinc ions, followed by binding of moxalactam and cleavage of the β -lactam structure.

While the ten structures help to provide high-resolution information about the catalytic events in the active site of L1 MBL, the data presented in the manuscript does not provide novel mechanistic insight to advance our understanding of the MBL family of enzymes.

This is because, as the authors stated in the Introduction section, the catalytic mechanism for MBLs has been studied extensively by x-ray crystallography, spectroscopy, SAXS and computational modeling, including previous work by the authors themselves. As a result, most steps of the β -lactam hydrolysis process by MBL are well characterized, except for those short-lived events such as activation of a catalytic water molecule and the subsequent nucleophilic attack on the β -lactam substrate.

It would have been great if this manuscript, with its ultrafast TR-SSX technology, could provide novel functional insight on these transient states that are not tractable by other experimental approaches. However, findings reported in this manuscript essentially conform to existing knowledge with little new information.

For example, the authors showed that the binding of two zinc ions to the active site precedes the binding of the β -lactam substrate. This particular order of events may be a result of the experimental setup, when zinc ions were withheld in a "cage" and only released by femtosecond laser pulse. Under in vivo conditions, zinc ions are believed to be stably associated with the active site of L1 MBL.

The authors did observe cleavage of the β -lactam ring at the time point of 150 ms post the laser pulse. While this observation is interesting, subsequent structural analysis by the authors to track inter-atom distances between the zinc ions and active site residues, yielded no mechanistic insight. The authors did propose two water molecules near the zinc ions, namely W423 and W470, as potential candidates of the "attacking" water molecule but such proposal should be further investigated by the authors by either computational or experimental approaches.

Overall, this manuscript reports findings that conform to existing knowledge in the field but lack novel mechanistic insights to inspire further studies.

Reviewer #2 (Remarks to the Author):

The reaction mechanism of metallo-beta-lactamases (MBLs) has been the subject of many recent works, that identified the accumulation of reaction intermediates for the hydrolysis of carbapenems and cephalosporins. However, the early steps of the reaction (formation of the Michaelis complex) have remained partially elusive. In this work, Joachimiak and

coworkers report the use of time-resolved serial synchrotron crystallography to characterize the first 4 seconds of the hydrolysis of the oxa-cephem moxalactam by the B3 MBL L1, being able to capture events during the first milliseconds of the reaction. Key to the success of this work relies in the strategy for sample preparation, in which the (inactive) apo L1 is incubated with the substrate, and the enzyme activation is triggered by metal binding upon laser-induced release from a photo-labile caged zinc complex. This is innovative and allows the authors to provide a detailed structural description of the reaction.

The work has been competently performed by highly qualified researchers using a state-of-the-art approach, and the results are novel and of high interest to the community, in general. There are, however, several aspects that the authors should consider in the work to really have an impact in the field. The main weakness of the work is the apparent lack of knowledge of the authors of non-crystallographic previous mechanistic work on MBLs by several groups. Some previous observations are confirmed by their work, and this should be indicated, and other, more controversial aspects, should be addressed. Otherwise, the paper will fall short in providing a structural description of a biochemical relevant aspect without going deep in the biochemistry of the topic. In addition, the article needs a careful revision of the writing style, as well as of the references, that look like taken from a general database without paying attention to their content.

1. The authors present the finding of the first intermediate in which the nucleophilic attack and the C-N bond cleavage have already taken place. This intermediate was early proposed by Benkovic and Crowder for nitrocefin, and later by Vila for carbapenems, who later showed that these two steps were synchronous.

2. Most of the novelty of the work resides in the structural demonstration that these two steps are synchronous, and in revealing the binding mode of the substrate to the enzyme. This should be stressed more. It is also very important that the authors observe minimal changes at the active site, restricted to Zn-Zn distance changes while the whole protein structure experiences minor changes.

3. The choice of the couple L1-moxalactam was originally proposed by Spencer (ref 6) to exploit the fact that L1 hydrolyzes this antibiotic with a small K_M and a small k_{cat} , favoring trapping of the first steps. The authors do not mention this as a key element of the success of their experiment, supplemented by the novel strategy of zinc release in the crystal. Instead, they mention (P. 6) that L1 was selected because it can be purified in milligram quantities and produce well diffracting crystals. This is also true for many other MBLs, even for some more clinically relevant. This does not make justice to the original strategy from Spencer that is replicated here.

4. The role of Zn_2 for substrate binding has been early demonstrated, and it is not discussed (Rasia et al, J Biol Chem. 2004 Jun 18;279(25):26046-51).

5. The discussion regarding the alternative role of Asp120 is highly speculative (p.16), and not supported by the data. The authors should remove or tone down this section.

6. I could not have access to the Movie depicting the reaction in the Supplementary Material.

7. They suggest (p. 16) that protonation of N5 may be the rate limiting step. The Benkovic and Vila labs have already demonstrated this.

8. Figure S1. Which is the concentration of the enzyme? It is intriguing that there is scarce absorption at this range coming from the protein.

The authors should also revise some concepts well established in the field:

1. p. 3, second paragraph. When referring to lactamases, they mention "The enzymes are frequently very promiscuous (check spelling)....". This is not correct. Many SBLs are narrow spectrum enzymes, and B2 MBLs are exclusive carbapenemases. This statement could be applied to MBLs, but not to all lactamases.

2. p. 3, second paragraph. Reference to key mechanistic and structural work from different groups (Spencer, Crowder, Benkovic, Vila, Bonomo, Frere, Palzkill) is lacking here. A recent comprehensive review on MBLs is not considered (Bahr et al, Chem Rev. 2021 Jul 14;121(13):7957-8094).

3. p. 4, second paragraph. "L1 MBL shares close structural similarity of bi-metal ions architecture of the catalytic site with the NDM-1 from *K. pneumoniae* and other MBLs". This

is not correct, B3 enzymes (such as L1) are highly divergent from B1 enzymes (such as NDM-1), with a different metal site architecture, including different ligand sets and loop arrangements (Bahr et al, Chem Rev. 2021 Jul 14;121(13):7957-8094).

4. p. 4, third paragraph. "The catalytic mechanism for MBLs has been studied extensively". Some of these references do not even correspond to MBLs nor to mechanistic studies. The early work from Benkovic, and later work from Crowder and Vila is not cited here. Epidemiological reviews such as 11 do not belong here.

5. p. 4, third paragraph. Reference 14 does not provide the best QM/MM calculation. The authors should include the work from Nair and coworkers on NDM-1 (Das et al, Phys Chem Chem Phys. 2017 May 24;19(20):13111-13121; Tripathi et al, ACS Catal. 2015, 5, 4, 2577-2586). However, the most accepted hypothesis, and supported by work from the Page and Vila labs indicate that the attacking nucleophile is a zinc-bound hydroxide/water molecule. Same for referencing computational work in p. 9.

6. p. 4, third paragraph. "Proposed MBLs mechanisms indicate that H118 together with D120 act as a general base coordinating the water molecule activated by zinc center which initiates nucleophilic attack on the carbonyl carbon of β -lactam opening the ring." There are no references provided, and this is only true for B2 MBLs.

7. p. 4, third paragraph. The authors mention evidence from SAXS while crystallographic papers. Instead, work from Tierney should be discussed here using EXAFS and XANES (ref. 26, Lisa et al and Brece et al., J Am Chem Soc. 2009 Aug 26;131(33):11642-3.). This previous work should be discussed in page 10, when describing changes in the Zn-Zn distance. Same in page 12. SAXS is not EXAFS!!!!

8. Again, spectroscopic trapping of intermediates by Benkovic, Vila and Crowder is not discussed.

9. P. 7, last paragraph. "The active site of the L1 is located on the edges of β sheets and is surrounded by several loops. It possesses the most common active site motive for MBLs and comprises a sequence motif H116-X-H118-X-D120-H121". This is again incorrect and misleading, as mentioned before.

10. P, 17. "The partial occupancy for the second zinc can help explain subclass B2 active with one zinc, but the mechanism may involved additional protein residues". This statement not only is completely wrong, but it also reveals the lack of knowledge of the field of MBLs by the authors. The partially depleted Zn site is the Zn2 site, which is actually the only one present in B2 enzymes. Moreover, B2 enzymes are mono-zinc since one of the His ligands from the Zn1 site is replaced by an Asn. They should correct this sentence. Studies on B2 enzymes date back to the previous century.

Reviewer #3 (Remarks to the Author):

Wilamowski et al

Manuscript#: NCOMMS-22-13268-T

Corresponding Author: Andrzej Joachimiak

Title: Time-Resolved β -lactam Cleavage by L1 Metallo- β -Lactamase

These are nice time-resolved crystallography results using pink beam from a synchrotron x-ray source, with pump-probe and serial MX strategies using fixed targets. They demonstrate formation of the ternary complex with L1 metallo-beta-lactamase from *Stenotrophomonas maltophilia*, two Zn ions, and the antibiotic substrate moxalactam for several time points (20, 40, 60, 80 and 100 ms) after the illumination pump. At the 150 ms timepoint and longer, the electron density is interpreted as a hydrolysed beta-lactam, but still bound to the dinuclear Zn centre. The methods and results will be of interest to the readers interested in time-resolved structural biology and in antimicrobial resistance. The ability to trap an enzyme-substrate complex in a metallo-beta-lactamases is rare and the use of EDTA to strip crystals of Zn, and then use pump-probe methods to release Zn(II)

from a caged Zn-compound is outstanding.

The paper suffers a bit from too much brevity to fully explain some of the results. It also has only modest resolution of the diffraction data. The authors present an interpretation of ten, time-resolved TR-SSX datasets based upon fitting atomic models to a limited sequence of events -- all with 100 % occupancy. This may not be the case in reality, and it is unclear if the quality of the data is sufficient to evaluate the potential for partial occupancies of different atomic models at the different time points.

Despite these shortfalls, this is a good paper and in the opinion of this reviewer, it should be published, provided the comments below are address. Of these comments, the first two are the most significant (Page 20 & 24 and Page 14 - Figure 4,) and must be addressed thoroughly.

Page 20 & 24

"... (Fig. 1). To avoid impact of de-caged zinc ions diffusing during the time delay through crystals on the ALEX chip, we used several step sizes during chip scan."

"We used a 60 μm grid made from nylon (NY6004700, Millipore) that hold 15 μL of crystals slurry sandwiched between two layers of 6 μm mylar."

How do the authors know that de-caging occurs only within the optical laser spot size? Indeed, these pump-probe, time-resolved serial MX studies conducted at room temp used a fixed target that is essentially entirely photo transparent. There is little text nor discussion of how the authors determined that the pump-probe scheme did not liberate Zn atoms beyond the focus of the optical laser spot due to scattering to other regions of the chip. For instance, were any dark data interleaved and/or taken from one or more chips that were used for pump-probe datasets? It appears that all the dark data came from unique chip(s), and all the pump-probe data come from illuminated chips, but they were never mixed or interleaved. For example, a row of dark data collected between two illumination rows, etc. The time-point datasets do appear to be internally consistent with each other, but optical laser light scattering, and commensurate systematic de-caging might be a systematic problem. The authors should include more information addressing this concern in the methods and/or supplemental information sections.

Page 14. Figure 4, and surrounding text & methods; as well as Supplementary Figure S4 "Figure 4. TR-SSX crystal structures of moxalactam of the active site of L1 MBL."

The interpretation of TR-SSX datasets can be challenging, especially if there are mixtures of species and/or differing occupancies. The modest resolution of these structures also makes interpretation of alternative conformations difficult, and probably not well supported in this case. However, for the 100 and 150 ms time point structures, it will be essential to show electron density maps for alternative atomic models. For instance, how does the 100 ms TR-SSX data refine against a model of the hydrolysed product? Similarly, how does the 150 ms time point TR-SSX data refine against an authentic substrate atomic models? What do the difference maps for these two alternatives look like. What have the authors done to conclude that these datasets are best modelled with pure atomic models of either substrate or product? Is it not possible that there is a mixture of substrate and product in these two datasets?

What do the isomorphous difference maps ($F_o - F_c$) look like when comparing different timepoints?

For the TR-SSX datasets, the number of crystal lattices merged is very small (Supplementary Table S1A), which leads to poor stats in the highest resolution shell. The authors should add information on the minimum number of lattices needed for this space group and the pink beam conditions used to collect the data. As it reads now, the modest resolution and quality of the electron density maps are deleteriously impacted by the low

number of lattices and poor statistics in the highest resolution shells, which reduces the potential impact of the results.

Additional comments follow from here.

Page 2 - abstract

... we showed the time course of β -lactam hydrolysis and assembled molecular movie spanning 4 seconds."

The use of the term "molecular movie" is very misleading since movies typically run at 24 - 60 frames per second. It is far more accurate to state that they present the dark state and a stop-motion sequence of ten time-resolved atomic models from 20 ms after laser illumination through 4 seconds.

Or, to state explicitly and more preferably in the abstract that they collected discrete time-points across a "logarithmic time scale" at 20, 40, 60, 80, 100, 150, 300, 500, 2000, and 4000 ms after laser illumination and before exposure to "pink" beam x-ray photons of either 3.7 μ s or 7.4 μ s duration.

In the abstract, please indicate the typical resolution observed for the time-resolved data.

"... bound to L1 metallo- β -lactamase (MBL)." Is too ambiguous, please indicate the source of the enzyme.

Page 6

"...provide opportunity to uncover details of binding events, and subsequent..."

Please correct the spelling error.

Page 7

"... most common active site motive for MBLs and comprises a sequence..."

The authors probably mean "motif"

Page 9

"Computational modeling of the cleavage reaction progression in solution shows that β -lactam ring hydrolysis is completed within 2000 ms (ref 15)."

The reader may wonder why the authors rely upon computational estimates and not analysis of transient kinetic data of the actual reaction? It is common for enzyme reaction mechanisms to have fast and slow steps, and that product release is often a rate limiting step. Moreover, the TR-SSX results presented here suggest that once the correct enzyme-substrate complex is realized, then the hydrolysis reaction happens faster than 50 ms. Indeed, comparison of the atomic models for 100 ms and 150 ms time-points suggests that 100% E-S complex is converted to 100% hydrolysis product within the 50 ms equilibration time between time-points. Therefore, one might assume that the hydrolysis reaction must be significantly faster than 50 ms. The computational study might address the time needed for the hydrolysis step, but the authors only reference that reaction is complete within 2000 ms.

Page 10 - dark structures soaked with caged Zn and moxalactam

"In this structure we observe partly occupied zinc ion in Zn2 site, suggesting that the metal cage was somewhat leaky, however this structure does not contain the electron density for

moxalactam (Fig. 3b)."

Please state if the [Zn(XDPAdCage)]⁺ caged compound is visible/ordered in the dark structure and if so, where was it with respect to the four active sites in the homotetramer L1 enzyme? If no caged compound is observed in the dark state, then please state explicitly that it is disordered in the crystals. There is also a lack of information regarding the crystal packing, overall solvent content, and lattice channels size(s) that may help the reader better understand or estimate how far the Zn atoms and moxalactam must diffuse to reach the active site. The reader will also wonder if moxalactam is observed in the dark state.

This reviewer took the time to look at the symmetry and crystal lattice packing of 6uac (the PDB atomic model used for molecular replacement in this study). Of importance to this study is the near isomorphous unit cell dimensions (Space Group: P 6₄ 2 2; a = b = 104.34 Å, c = 98.98 Å), and noted the very easy access of each active site to a central cavity measuring about a 50 Å diameter that runs along the 6(4) screw symmetry axis and throughout the entire crystal lattice. Thus, it is easy to envision that both the caged Zn compound and the moxalactam could be disordered throughout the lattice and yet still be remarkably close to an active site. The authors should indicate if the largest dimensions of the [Zn(XDPAdCage)]⁺ caged compound and the moxalactam substrate are in fact smaller than the diameter of the 6(4) screw channels that traverses the crystal lattice. In the reviewer's opinion, an illustration of the solvent channel and relationship to the active sites is worth including in the manuscript.

Page 10

"... It displaces several water molecules"

Ambiguous, does "It" refer to one or both of the Zn(II) ions, or to the moxalactam?

Page 11

"Figure 3. TR-SSX structures of moxalactam bound to the active site of L1 MBL."

Part "b" of the figure is far less useful than parts "a" and "c", the latter two are both essential. Part "b" is also less useful than Supplementary Figure S4, since the latter shows electron density maps too. There are almost no changes apparent in the (electron density, Fig S4, and) atomic models for the 20 – 100 ms structures, and so they could be combined. Like the criticism above, it would be more useful to show features of electron density difference maps that support why the 100 ms timepoint dataset is interpreted as substrate, but the 150 ms is refined against a product atomic model. An important example is to show isomorphous Fo-Fo difference maps for different time point datasets.

Page 13 -- Figure 4

"a. L1 active site structure at 150 ms..."

"b. Electron density..."

"f. TR "pink" beam structures..."

Please add labels for a few of the key residues. It is also wise to include interactions between the Zn atoms and their coordinating atoms to help illustrate coordination sphere.

"g. Catalytic mechanism proposed for L1 MBL based on TR-SSX experiments."

It would be good to indicate that these "intermediate" structures are not observed in the time-resolved atomic models and apparently happen within the 50 ms between the 100 and 150 ms time points. It would also be good to indicate if the activated water/hydroxide is either the bridging solvent or a terminal ligand to one of the Zn atoms.

The coordination of the solvent molecule(s), protein ligands, and substrate atoms to the Zn ions are not indicated. This makes it more difficult to figure out where the proposed nucleophilic solvent is "created" and from what direction it attacks the substrate bond to be broken. In principle, this is the type of information that could/should come from a time-resolved study. However, it appears that in these results the modest resolution and/or dynamics of the reaction cloak the solvent atoms to crystallographic analysis and yield less than obvious electron density maps.

In addition, the detailed mechanistic paragraphs (from the last paragraph on page 13 through middle of page 16) should include a better figure for the proposed reaction mechanism than shown in part "g". An improved figure will also help with the discussion text, which also needs to be sharpened and a bit more focused. For instance, at the page 15-16 interface the authors state, "A water molecule can be attracted by di-zinc center, and it becomes activated "attacking" water and it rapidly reacts with C-8 that is precisely positioned for reaction." Here the text is very poorly worded; this matters, and an "activated water" is probably better described as a hydroxide ion (as stated a few lines lower), not a water molecule. The combination of a poor mechanism figure and wandering text significantly decreases the impact of results and insights gleaned from the TR-SSX structures.

Page 18

"...we indexed, and integrated 5191 (3374) diffraction images per data set."

Throughout, it would be clearer to indicate the number of crystal lattices merged per dataset since some images may or may not contain more than one lattice. Despite many thousands of images with apparent diffraction spots (e.g. table of data collection and refinement stats), there are only a few hundreds of crystal lattices actually merged. Why do so many images exhibit strong spots, but apparently fail to integrate and/or merge into the whole TR-SSX dataset?

Page 18

"... polychromatic x-rays (1.02-1.18 Å wavelength range) were focused using Kirkpatrick-Baez mirrors..."

Please also add values in eV too, and indicate the % band pass for these pink beam studies.

Page 20

"We used nanosecond pulses from an OPOTEK Opolette 355 II HE laser, focused to 100 x 80 μm spot."

More detail is needed for the photo-physics of the de-caging reactions. How many nanosecond pulses for what total illumination time were used? What was the optical light intensity/power for the de-caging reaction? What is the quantum efficiency given the illumination and sample conditions?

Page 24

"Supplementary Figure S1. Photolysis of [Zn(XDPAdCage)]⁺ with 347 nm UV light in solution triggers cleavage of meropenem in a presence of L1 from *S. maltophilia*. Prior experiment zinc ions were removed from L1 by dialysis against buffer B containing EDTA."

More information is needed, please. Is this analysis for enzyme in solution or in a single crystal, or a crystal slurry? How much EDTA for how long and at what temperature, pH? It is very difficult to tell if these spectra differ from spectra shown in ref 32 (J. Am. Chem. Soc. 141, 12100–12108 (2019)). The text suggests that there is a change in the optical

spectrum upon hydrolysis of moxalactam; however the caged Zn compound and photoproducts are also coloured and contribute to the observed spectra. It would be good to help the reader deconvolute the optical spectra.

Page 24

"Final concentration of moxalactam was 6.58 mM and the concentration of [Zn(XDPAdCage)]⁺ was 3.29 mM."

Please provide the reader with an estimate for the enzyme concentration (and the active site metal concentration) within the crystals. The reader may note that the substrate concentration is twice that of the Zn, but each active site binds two Zn atoms and only one antibiotic. Please provide a K_d and/or K_m for L1 and moxalactam. One may assume that the crystals are several mM enzyme, and therefore wonder if this is enough metal to fully occupy the active sites throughout the crystal. The reader will also wonder what the quantum efficiency of the de-caging process is; please provide a brief summary of the photo-physics characteristics.

Page 25

"Supplementary Movie S1. Snapshots of moxalactam cleavage by L1 from *S. maltophilia* captured by TR-SSX (20 – 4000 ms)."

The reviewer did not have access to Movie S1. Is the legend detailed enough so that the reader understands what is being shown? Are the transitions between time-points molecular morphs or does it show authentic steps without morphs between them. Is there electron density shown, and if so, then to what resolution and what type of maps...

Page 26

"Supplementary Figure S1. Photolysis of [Zn(XDPAdCage)]⁺ with 347 nm UV light in solution triggers cleavage of meropenem in a presence of L1 from *S. maltophilia*. Prior experiment zinc ions were removed from L1 by dialysis against buffer B containing EDTA."

This poorly worded legend also needs significantly more detail. It is not at all clear what the figure shows. Does "NO" stand for a dark sample/control? What is /are the concentration(s) of the [Zn(XDPAdCage)]⁺ used in these experiments, temperature of the reaction?, illumination power?, other reagents in the crystal slurry or solution(s)?, How much starting meropenem is in the reactions? How much meropenem is cleaved during this time? Which lambda max goes with which species? How does this compare to Figure 4 in ref 32 (J. Am. Chem. Soc. 141, 12100–12108 (2019)) – perhaps add another panel to show this comparison?

Page 29

"Supplementary Figure S4. Evolution of electron density in the L1 MBL active site during TR-SSX experiments."

More information is needed for the legends. For example, are these 2 Fo-Fc maps carved around residues of interest and contoured at 1 sigma (and to what resolution)? Are these typical omit maps, or are they Polder omit maps?

Page 30

"Supplementary Figure S5. ... (red – high B-factor; dark blue – low B-factor)."

Please be quantitative, provide the values

Page 31

“Supplementary Table S1A. SSX data collection and processing statistic of L1 β -lactamase crystals.”

The reader will wonder what are the estimated average x-ray doses to the crystals (expressed in Gy) used for these studies; therefore, please provide this information.

What are the average B-values for the antibiotic and the Zn atoms in the TR-SSX datasets. Should the reader assume 100% occupancy for these ligands in each timepoint dataset?

REVIEWER COMMENTS

Reviewer #1 (Remarks to the Author):

Q. Metallo- β -lactamases (MBLs) play crucial roles in conferring antibiotic resistance to pathogenic bacteria. Understanding the molecular mechanism of how these enzymes hydrolyze existing β -lactam antibiotics is of critical importance to drug discovery effort to develop novel anti-bacterial therapeutics.

The authors used time-resolved x-ray crystallography to track the hydrolysis of moxalactam in the active site of L1, an emerging MBL from the opportunistic gram-negative basillus pathogen *S. maltophilia*. Their time resolved synchrotron light source (TR-SSX) system is state-of-the-art, capable of tracking the catalytic process over the duration of 5-4000 millisecond. Altogether, ten structures representing the L1 enzyme at different time points of hydrolyzing moxalactam were obtained. These snapshots capture distinct steps of the catalytic process, including binding of the zinc ions, followed by binding of moxalactam and cleavage of the β -lactam structure.

While the ten structures help to provide high-resolution information about the catalytic events in the active site of L1 MBL, the data presented in the manuscript does not provide novel mechanistic insight to advance our understanding of the MBL family of enzymes.

This is because, as the authors stated in the Introduction section, the catalytic mechanism for MBLs has been studied extensively by x-ray crystallography, spectroscopy, SAXS and computational modeling, including previous work by the authors themselves. As a result, most steps of the β -lactam hydrolysis process by MBL are well characterized, except for those short-lived events such as activation of a catalytic water molecule and the subsequent nucleophilic attack on the β -lactam substrate.

It would have been great if this manuscript, with its ultrafast TR-SSX technology, could provide novel functional insight on these transient states that are not tractable by other experimental approaches. However, findings reported in this manuscript essentially conform to existing knowledge with little new information.

For example, the authors showed that the binding of two zinc ions to the active site precedes the binding of the β -lactam substrate. This particular order of events may be a result of the experimental setup, when zinc ions were withheld in a “cage” and only released by femtosecond laser pulse. Under in vivo conditions, zinc ions are believed to be stably associated with the active site of L1 MBL.

The authors did observe cleavage of the β -lactam ring at the time point of 150 ms post the laser pulse. While this observation is interesting, subsequent structural analysis by the authors to track inter-atom distances between the zinc ions and active site residues, yielded no mechanistic insight. The authors did propose two water molecules near the zinc ions, namely W423 and W470, as potential candidates of the “attacking” water molecule but such proposal should be further investigated by the authors by either computational or experimental approaches.

Overall, this manuscript reports findings that conform to existing knowledge in the field but lack novel mechanistic insights to inspire further studies.

Response: Yes we have confirmed the proposed general mechanism but we have also eliminated some alternative proposed mechanisms including involvement of bridging water molecule. As was discussed

recently (Bahr et al. 2021), there are still several controversial issues in the field of MBLs, including: are catalytic events synchronous or occur in discrete steps with accumulation of reaction intermediates, the identity intermediates, what molecule serves as the nucleophile and the proton donor, what is the rate-determining step of the reaction and the role and essentiality of each zinc ions in the mechanism. We clearly were able to address the essentiality of both zinc ions in substrate binding and in catalysis. For the first time we have shown that the enzyme holds the substrate as it adapts in the catalytic site without reaction occurring. Once substrate adjusts to the proper conformation the reaction progresses very fast. An incoming water molecule is activated by di-zinc center and serves as nucleophile. Our data suggest more dynamic mechanism with ligand conformational changes playing much more significant role than the enzyme conformational adjustments.

Reviewer #2 (Remarks to the Author):

Q. The reaction mechanism of metallo-beta-lactamases (MBLs) has been the subject of many recent works, that identified the accumulation of reaction intermediates for the hydrolysis of carbapenems and cephalosporins. However, the early steps of the reaction (formation of the Michaelis complex) have remained partially elusive. In this work, Joachimiak and coworkers report the use of time-resolved serial synchrotron crystallography to characterize the first 4 seconds of the hydrolysis of the oxa-cephem moxalactam by the B3 MBL L1, being able to capture events during the first milliseconds of the reaction. Key to the success of this work relies in the strategy for sample preparation, in which the (inactive) apo L1 is incubated with the substrate, and the enzyme activation is triggered by metal binding upon laser-induced release from a photo-labile caged zinc complex. This is innovative and allows the authors to provide a detailed structural description of the reaction.

The work has been competently performed by highly qualified researchers using a state-of-the-art approach, and the results are novel and of high interest to the community, in general. There are, however, several aspects that the authors should consider in the work to really have an impact in the field. The main weakness of the work is the apparent lack of knowledge of the authors of non-crystallographic previous mechanistic work on MBLs by several groups. Some previous observations are confirmed by their work, and this should be indicated, and other, more controversial aspects, should be addressed. Otherwise, the paper will fall short in providing a structural description of a biochemical relevant aspect without going deep in the biochemistry of the topic. In addition, the article need a careful revision of the writing style, as well as of the references, that look like taken from a general database without paying attention to their content.

Response: We thank reviewer for pointing to earlier work and in the revised manuscript we have included numerous earlier references that describe the b-lactamase mechanism.

Q1. The authors present the finding of the first intermediate in which the nucleophilic attack and the C-N bond cleavage have already taken place. This intermediate was early proposed by Benkovic and Crowder for nitrocefin, and later by Vila for carbapenems, who later showed that these two steps were synchronous.

Response: Yes, it could be possible that these two steps are synchronous but our data, without an image between 100 ms and 150 ms, may not be able to support directly either way. We have elaborated on this subject in the revised manuscript and added references.

Q2. Most of the novelty of the work resides in the structural demonstration that these two steps are synchronous, and in revealing the binding mode of the substrate to the enzyme. This should be stressed

more. It is also very important that the authors observe minimal changes at the active site, restricted to Zn-Zn distance changes while the whole protein structure experiences minor changes.

Response: Yes, revised manuscript to reflect reviewer's comments.

Q3. The choice of the couple L1-moxalactam was originally proposed by Spencer (ref 6) to exploit the fact that L1 hydrolyzes this antibiotic with a small KM and a small kcat, favoring trapping of the first steps. The authors do not mention this as a key element of the success of their experiment, supplemented by the novel strategy of zinc release in the crystal. Instead, they mention (P. 6) that L1 was selected because it can be purified in milligram quantities and produce well diffracting crystals. This is also true for many other MBLs, even for some more clinically relevant. This does not make justice to the original strategy from Spencer that is replicated here.

Response: Yes, we make justice to the original strategy proposed by Spencer that is replicated here. However, high quality of protein and crystals was very important factor to assure reproducibility of the data and sufficient resolution that allows us to interpret electron density.

Q4. The role of Zn²⁺ for substrate binding has been early demonstrated, and it is not discussed (Rasia et al, J Biol Chem. 2004 Jun 18;279(25):26046-51).

Response: Yes, we discuss observations by Rasia & Vila that are confirmed by our time resolved experiments.

Q5. The discussion regarding the alternative role of Asp120 is highly speculative (p.16), and not supported by the data. The authors should remove or tone down this section.

Response: Yes, we have toned down this section.

Q6. I could not have access to the Movie depicting the reaction in the Supplementary Material.

Response: We are sorry that file with molecular visualization of the chemical transformation was not active. This is corrected in the revised manuscript. We have also included another file in Supplemental Movie 1 that shows evolution of electron density over time.

Q7. They suggest (p. 16) that protonation of N5 may be the rate limiting step. The Benkovic and Vila labs have already demonstrated this.

Response: Yes, and we cited these references.

Q8. Figure S1. Which is the concentration of the enzyme? It is intriguing that there is scarce absorption at this range coming from the protein.

Response: Yes, protein concentration is very low (0.1 μ M) and very little contributes to spectra of the meropenem, but zinc cage has some contribution, particularly in UV range. We have now included in Supplemental Figure S1, figure S1b that shows contribution of zinc cage to spectrum of meropenem at two different concentrations of the cage.

Q. The authors should also revise some concepts well established in the field:

1. p. 3, second paragraph. When referring to lactamases, they mention “The enzymes are frequently very promiscuous (check spelling)...”. This is not correct. Many SBLs are narrow spectrum enzymes, and B2 MBLs are exclusive carbapenemases. This statement could be applied to MBLs, but not to all lactamases.

Response: Yes, we agree many SBLs are narrow spectrum, at the same time many can hydrolyze variety of substrates. They belong to two very large superfamilies with large diversity of substrates. We have determined many structures of beta lactamases including SaxA (isothiocyanate hydrolase) that is MBL and can hydrolyze multiple substrates. However, as suggested by reviewer we corrected this statement and spelling.

Q2. p. 3, second paragraph. Reference to key mechanistic and structural work from different groups (Spencer, Crowder, Benkovic, Vila, Bonomo, Frere, Palzkill) is lacking here. A recent comprehensive review on MBLs is not considered (Bahr et al, Chem Rev. 2021 Jul 14;121(13):7957-8094).

Response: We updated references as suggested by reviewer.

Q3. p. 4, second paragraph. “L1 MBL shares close structural similarity of bi-metal ions architecture of the catalytic site with the NDM-1 from *K. pneumoniae* and other MBLs”. This is not correct, B3 enzymes (such as L1) are highly divergent from B1 enzymes (such as NDM-1), with a different metal site architecture, including different ligand sets and loop arrangements (Bahr et al, Chem Rev. 2021 Jul 14;121(13):7957-8094).

Response: Although as pointed out by reviewer B3 enzymes are divergent from B1 enzymes and have different ligand sets and loop arrangements the active sites and metal coordination of NDM-1 (B1) and L-1 (B3) are very similar (please see figures below). Here we compare atomic resolution structure of NDM-1 active site (4hl2) with bound hydrolyzed ampicillin (green carbons) and 1.52 Å L1 MBL in complex with hydrolyzed moxalactam (6U13) (magenta carbons). Five residues coordinating di-zinc center are identical and their conformation is similar. The only difference is replacement of NDM-1 Cys208 residue with His110 in L1 MBL. Moreover, the interaction of the products carboxylates with di-zinc center is very similar. We have adjusted text in the revised manuscript to address reviewer comments and cite Bahr et al reference). Active site similarity of B1, B2 and B3 MBLs is underscored by recently published inhibitor (Hinchliffe et al. Inf. Dis., 2021, PMID: 34355567) that is capable to inhibit all three classes of MBLs by direct interaction with the mono- or dizinc centers.

Q4. p. 4, third paragraph. “The catalytic mechanism for MBLs has been studied extensively”. Some of these references do not even correspond to MBLs nor to mechanistic studies. The early work from Benkovic, and later work from Crowder and Vila is not cited here. Epidemiological reviews such as 11 do not belong here.

Response: The references were updated as appropriate in the text.

Q5. p. 4, third paragraph. Reference 14 does not provide the best QM/MM calculation. The authors should include the work from Nair and coworkers on NDM-1 (Das et al, Phys Chem Chem Phys. 2017 May 24;19(20):13111-13121; Tripathi et al, ACS Catal. 2015, 5, 4, 2577–2586). However, the most accepted hypothesis, and supported by work from the Page and Vila labs indicate that the attacking nucleophile is a zinc-bound hydroxide/water molecule. Same for referencing computational work in p. 9.

Response: We included short discussion of NDM-1 and references suggested by reviewer.

Q6. p. 4, third paragraph. “Proposed MBLs mechanisms indicate that H118 together with D120 act as a general base coordinating the water molecule activated by zinc center which initiates nucleophilic attack on the carbonyl carbon of β -lactam opening the ring.” There are no references provided, and this is only true for B2 MBLs.

Response: D120 is conserved in B-type MBLs but its role is still not fully understood. As discussed by Bahr et al. the main role of D120 (and H118) is to adequately position Zn²⁺ for substrate binding and catalysis and we changed text accordingly. We change the sentence as follow.

“However, nucleophile activation by metal ions does not require a general base to deprotonate the water molecule. Therefore, the role of D120 together with conserved residue H118 is to position Zn²⁺ for substrate binding and catalysis. The incoming water (for example W423) is activated and water/hydroxide attacks C-8 and forms intermediate anion(s) (**Fig. 4g**). Ring cleavage is completed through a proton translocation process that possibly utilizes another bulk water molecule as it was suggested previously¹⁴. The protonation of the N-5 nitrogen atom may be the limiting step. In 100 ms structure there are two water molecules that are closest to zinc ions (W423 and W470) and they may be good candidates for the reaction.”

Q7. p. 4, third paragraph. The authors mention evidence from SAXS while crystallographic papers. Instead, work from Tierney should be discussed here using EXAFS and XANES (ref. 26, Lisa et al and Breece et al., J Am Chem Soc. 2009 Aug 26;131(33):11642-3.). This previous work should be discussed in page 10, when describing changes in the Zn-Zn distance. Same in page 12. SAXS is not EXAFS!!!!

Response: Yes, we know difference between EXAFS and XANES and SAXS and we provide correct references (Lisa et al PMID: 28912448 and Breece et al PMID: 19653676).

Q8. Again, spectroscopic trapping of intermediates by Benkovic, Vila and Crowder is not discussed.

Response: We discuss spectroscopic trapping of intermediates of MBLs using EPR and Raman spectroscopies that help to understand mechanism and included references.

Q9. P. 7, last paragraph. “The active site of the L1 is located on the edges of β sheets and is surrounded

by several loops. It possesses the most common active site motive for MBLs and comprises a sequence motif H116-X-H118-X-D120-H121". This is again incorrect and misleading, as mentioned before.

Response: Here we disagree with reviewer, the active sites of MBLs is highly conserved in sequence and also structurally. My lab has determined over 100 structures of different lactamases with over 20 structures for both NDM-1 and L1 metallo-beta-lactamases. Some of these structures were determined at atomic resolution. The active sites of these enzymes are highly conserved. We believe the differences in catalytic properties and substrate binding of these enzymes are contributed by other components of the active sites (substrate recognition residues, loops, solvent structure and dynamics). We believe our observations are relevant to other di-metal beta-lactamases.

Q10. P, 17. "The partial occupancy for the second zinc can help explain subclass B2 active with one zinc, but the mechanism may involve additional protein residues". This statement not only is completely wrong, but it also reveals the lack of knowledge of the field of MBLs by the authors. The partially depleted Zn site is the Zn2 site, which is actually the only one present in B2 enzymes. Moreover, B2 enzymes are mono-zinc since one of the His ligands from the Zn1 site is replaced by an Asn. They should correct this sentence. Studies on B2 enzymes date back to the previous century.

Response: We corrected this sentence as suggested by reviewer.

Reviewer #3 (Remarks to the Author):

Wilamowski et al

Manuscript#: NCOMMS-22-13268-T

Corresponding Author: Andrzej Joachimiak

Title: Time-Resolved β -lactam Cleavage by L1 Metallo- β -Lactamase

These are nice time-resolved crystallography results using pink beam from a synchrotron x-ray source, with pump-probe and serial MX strategies using fixed targets. They demonstrate formation of the ternary complex with L1 metallo-beta-lactamase from *Stenotrophomonas maltophilia*, two Zn ions, and the antibiotic substrate moxalactam for several time points (20, 40, 60, 80 and 100 ms) after the illumination pump. At the 150 ms timepoint and longer, the electron density is interpreted as a hydrolysed beta-lactam, but still bound to the dinuclear Zn centre. The methods and results will be of interest to the readers interested in time-resolved structural biology and in antimicrobial resistance. The ability to trap an enzyme-substrate complex in a metallo-beta-lactamases is rare and the use of EDTA to

strip crystals of Zn, and then use pump-probe methods to release Zn(II) from a caged Zn-compound is outstanding.

The paper suffers a bit from too much brevity to fully explain some of the results. It also has only modest resolution of the diffraction data. The authors present an interpretation of ten, time-resolved TR-SSX datasets based upon fitting atomic models to a limited sequence of events -- all with 100 % occupancy. This may not be the case in reality, and it is unclear if the quality of the data is sufficient to evaluate the potential for partial occupancies of different atomic models at the different time points.

Despite these shortfalls, this is a good paper and in the opinion of this reviewer, it should be published, provided the comments below are address. Of these comments, the first two are the most significant (Page 20 & 24 and Page 14 - Figure 4,) and must be addressed thoroughly.

Q. Page 20 & 24

"... (Fig. 1). To avoid impact of de-caged zinc ions diffusing during the time delay through crystals on the ALEX chip, we used several step sizes during chip scan."

"We used a 60 μm grid made from nylon (NY6004700, Millipore) that hold 15 μL of crystals slurry sandwiched between two layers of 6 μm mylar."

Response: Here we show how the scan was performed. We include this schematic as a Supplementary Figure S2.

Supplementary Figure S2. Scanning of the *ALEX* mesh holder for SSX data collection. Simplified depiction of step sizes between data collection positions for L1 MBL crystals at 14-ID beamline. **A.** Movement of the chip across the Y-axis with step size labeled as blue bars with an overlaid red baseline that indicates the diameter of a laser pulse spot. **B.** Scheme of fixed-target SSX data collection on nylon mesh crystals holder utilizing the pump-probe system. The X and Y axis step sizes of the chip are labeled as red arrows. SSX data collection were conducted as pump-probe system with specific time-delay between laser pulse and a train of 100 ps pink-beam X-ray pulses. To avoid the impact of zinc diffusion on data collection the step size of the SSX crystal holder was increased for the detection of longer time-points of moxalactam cleavage reaction. The construction of the *ALEX* crystal holder (submitted for publication), which contains partially separated compartments in a nylon grid is an advantage for the limitation of zinc diffusion after sample illumination during a pump-probe data collection. The UV laser size has been 100 μm and doesn't change over the experiment. The shortest step sizes were at 20 ms as the SSX chip moved of 200 μm and 250 μm along X and Y axes, respectively.

Q. How do the authors know that de-caging occurs only within the optical laser spot size? Indeed, these pump-probe, time-resolved serial MX studies conducted at room temp used a fixed target that is essentially entirely photo transparent. There is little text nor discussion of how the authors determined that the pump-probe scheme did not liberate Zn atoms beyond the focus of the optical laser spot due to scattering to other regions of the chip. For instance, were any dark data interleaved and/or taken from one or more chips that were used for pump-probe datasets? It appears that all the dark data came from unique chip(s), and all the pump-probe data come from illuminated chips, but they were never mixed or interleaved. For example, a row of dark data collected between two illumination rows, etc. The time-point datasets do appear to be internally consistent with each other, but optical laser light scattering, and commensurate systematic de-caging might be a systematic problem. The authors should include more information addressing this concern in the methods and/or supplemental

information sections.

Response: The dark data were not interleaved with the light data on the same chip. They were collected using separate chips. Here is rationale. The ALEX chip is made of two layers of mylar foil (2x6 μm) and nylon mesh (60 μm) plus protein crystals (~20-30 μm) and is quite transparent for the laser beam at 360 nm. The path for the light is short, less than 100 μm , and any scattering along direction of the chip should be low. The quantum yields for zinc cage are good, but ~75% of the photons aren't accomplishing uncaging. So, if there is any spill over, its uncaging efficiency will be very low. In addition, in the experiment we used excess of cage to trap any released metal. Therefore, if there is any light spill over outside focus region released zinc will be trapped by empty cages (femtomolar affinity zinc Zn ions) and broken cages which still have picomolar affinity. Moreover, we conservatively designed steps between points to be large (please see Supplemental Figure S2) and steps increased for the longer time delay points so zinc ions would not be able to reach crystal in the next laser focus area. Consistent with this is the fact that we do not observe and inconsistencies in the data and therefore our conclusions are valid.

Q. Page 14. Figure 4, and surrounding text & methods; as well as Supplementary Figure S4
"Figure 4. TR-SSX crystal structures of moxalactam of the active site of L1 MBL."

The interpretation of TR-SSX datasets can be challenging, especially if there are mixtures of species and/or differing occupancies. The modest resolution of these structures also makes interpretation of alternative conformations difficult, and probably not well supported in this case. However, for the 100 and 150 ms time point structures, it will be essential to show electron density maps for alternative atomic models. For instance, how does the 100 ms TR-SSX data refine against a model of the hydrolyzed product? Similarly, how does the 150 ms time point TR-SSX data refine against an authentic substrate atomic models? What do the difference maps for these two alternatives look like. What have the authors done to conclude that these datasets are best modelled with pure atomic models of either substrate or product? Is it not possible that there is a mixture of substrate and product in these two datasets?

Response: We agree with this reviewer that interpretation on TR-SSX is challenging, and we spent a lot of time on interpretation and refinement using very different approaches. To our surprise the substrate is bound to the active site for almost 100 ms and very little happens. These structures are consistent with uncut substrate. At 150 ms time point and there after the electron density is consistent with a product and refinement of mixture of states does not go well. This was also a surprise. But then we realized that diffusing of water molecule, activation and cleavage of beta-lactam bond is very fast (few ms) and in 50 ms can occur in majority of unit cells of the crystal. We have repeated TR-SSX experiments twice and observed this some effect. We have calculated difference electron density maps and these maps are consistent with uncut moxalactam at 20, 40, 60, 80 and 100 ms and fully cut substrate at 150, 300, 500, 2000 and 4000 ms. We have included this difference map calculated as $F_{\text{obs}}(150\text{ms}) - F_{\text{obs}}(100\text{ms})$ in the revised manuscript as Figure 4b. These maps are consistent with uncut moxalactam at 100 ms and cut at 150 ms.

Q. What do the isomorphous difference maps ($F_o - F_c$) look like when comparing different timepoints?

For the TR-SSX datasets, the number of crystal lattices merged is very small (Supplementary Table S1A), which leads to poor stats in the highest resolution shell. The authors should add information on the minimum number of lattices needed for this space group and the pink beam conditions used to collect

the data. As it reads now, the modest resolution and quality of the electron density maps are deleteriously impacted by the low number of lattices and poor statistics in the highest resolution shells, which reduces the potential impact of the results.

Response: We have included this difference map calculated as Fobs(150ms)-Fobs(100ms) in the revised manuscript as Figure 4b.

Q. Page 2 - abstract

... we showed the time course of β -lactam hydrolysis and assembled molecular movie spanning 4 seconds.”

The use of the term “molecular movie” is very misleading since movies typically run at 24 - 60 frames per second. It is far more accurate to state that they present the dark state and a stop-motion sequence of ten time-resolved atomic models from 20 ms after laser illumination through 4 seconds.

Or, to state explicitly and more preferably in the abstract that they collected discrete time-points across a “logarithmic time scale” at 20, 40, 60, 80, 100, 150, 300, 500, 2000, and 4000 ms after laser illumination and before exposure to “pink” beam x-ray photons of either 3.7 μ s or 7.4 μ s duration.

In the abstract, please indicate the typical resolution observed for the time-resolved data.

“... bound to L1 metallo- β -lactamase (MBL).” Is too ambiguous, please indicate the source of the enzyme.

Response: We modified the abstract as requested by reviewer.

Q. Page 6

“...provide opportunity to uncover details of binding events, and subsequent...”

Please correct the spelling error.

Response: Done

Q. Page 7

“... most common active site motive for MBLs and comprises a sequence...”

The authors probably mean “motif”

Response: Yes, corrected

Q. Page 9

“Computational modeling of the cleavage reaction progression in solution shows that β -lactam ring hydrolysis is completed within 2000 ms (ref 15).”

The reader may wonder why the authors rely upon computational estimates and not analysis of transient kinetic data of the actual reaction? It is common for enzyme reaction mechanisms to have fast and slow steps, and that product release is often a rate limiting step. Moreover, the TR-SSX results

presented here suggest that once the correct enzyme-substrate complex is realized, then the hydrolysis reaction happens faster than 50 ms. Indeed, comparison of the atomic models for 100 ms and 150 ms time-points suggests that 100% E-S complex is converted to 100% hydrolysis product within the 50 ms equilibration time between time-points. Therefore, one might assume that the hydrolysis reaction must be significantly faster than 50 ms. The computational study might address the time needed for the hydrolysis step, but the authors only reference that reaction is complete within 2000 ms.

Response: Yes, we realized that diffusing of water molecule, activation and cleavage of beta-lactam bond is very fast (few ms) and in 50 ms can occur in majority of unit cells of the crystal. The rate of beta-lactam cleavage was reported by others and MD simulations were also performed, and we have done MD simulation for NDM-1. We have added relevant references to the manuscript.

Q. Page 10 – dark structures soaked with caged Zn and moxalactam

“In this structure we observe partly occupied zinc ion in Zn2 site, suggesting that the metal cage was somewhat leaky, however this structure does not contain the electron density for moxalactam (Fig. 3b).”

Please state if the [Zn(XDPAdCage)]⁺ caged compound is visible/ordered in the dark structure and if so, where was it with respect to the four active sites in the homotetramer L1 enzyme? If no caged compound is observed in the dark state, then please state explicitly that it is disordered in the crystals. There is also a lack of information regarding the crystal packing, overall solvent content, and lattice channels size(s) that may help the reader better understand or estimate how far the Zn atoms and moxalactam must diffuse to reach the active site. The reader will also wonder if moxalactam is observed in the dark state.

Response: The zinc cage and moxalactam are present in our dark structure but they are disordered. We have statement in the text:

“For the TR-SSX experiments the [Zn(XDPAdCage)]⁺ and moxalactam substrate were presoaked in protein crystals. These crystals produced “dark” data set (**Table S1**).”

Q. This reviewer took the time to look at the symmetry and crystal lattice packing of 6uac (the PDB atomic model used for molecular replacement in this study). Of importance to this study is the near isomorphic unit cell dimensions (Space Group: P 6₄ 2 2; a = b = 104.34 Å, c = 98.98 Å), and noted the very easy access of each active site to a central cavity measuring about a 50 Å diameter that runs along the 6(4) screw symmetry axis and throughout the entire crystal lattice. Thus, it is easy to envision that both the caged Zn compound and the moxalactam could be disordered throughout the lattice and yet still be remarkably close to an active site. The authors should indicate if the largest dimensions of the [Zn(XDPAdCage)]⁺ caged compound and the moxalactam substrate are in fact smaller than the diameter of the 6(4) screw channels that travers the crystal lattice. In the reviewer’s opinion, an illustration of the solvent channel and relationship to the active sites is worth including in the manuscript.

Response: We agree with reviewer comments. Crystal packing of L1 MBL at the P6422 space group reveals the hexagonal organization of molecules. Crystals of L1 MBL contain one monomer in the asymmetric unit. However, the active physiological assembly of L1 MBL is a tetramer. Symmetry mates of L1 crystal structure were depicted as grey sticks as next, the secondary structures for tetramer biological assembly were shown as cyan, and the moxalactam bound to L1 tetramer was shown as yellow. To illustrate the localization of all active sites in the crystal of L1 MBL the zinc atoms were

illustrated as purple spheres. Observed crystal packing reveals that active sites of L1 MBL are faced toward exposed solvent channels that possess a diameter of 55 Å. Open “honeycomb” architecture of L1 MBL crystals facilitates diffusion of compounds used for the soaking of crystals before initiation of TR reaction. We assume that both [Zn(XDPAdCage)]⁺ caged compound and moxalactam (13.5 Å at longest intraatomic distances) occupy crystal solvent channels before reaction. Therefore, after de-caging the zinc atoms and moxalactam sequentially binds to the active site. We added panel d to supplementary figure S2.

Q. Page 10

“... It displaces several water molecules”

Ambiguous, does "It" refer to one or both of the Zn(II) ions, or to the moxalactam?

Response: Water molecules are displaced by both zinc ions and moxalactam. We added text to sentence:

“In the first data point collected at 20 ms, two bound Zn²⁺ ions are observed and partial electron density for a moxalactam molecule is also detected (**Fig. S5**). Zinc ions and antibiotic displace several water molecules which were present in the “dark” structure.”

Q. Page 11

“Figure 3. TR-SSX structures of moxalactam bound to the active site of L1 MBL.”

Part “b” of the figure is far less useful than parts “a” and “c”, the latter two are both essential. Part “b” is also less useful than Supplementary Figure S4, since the latter shows electron density maps too. There are almost no changes apparent in the (electron density, Fig S4, and) atomic models for the 20 – 100 ms structures, and so they could be combined. Like the criticism above, it would be more useful to show features of electron density difference maps that support why the 100 ms timepoint dataset is interpreted as substrate, but the 150 ms is refined against a product atomic model. An important example is to show isomorphous Fo-Fo difference maps for different time point datasets.

Response: Very important point, we have changed Figure 3 and included evolution of electron density around moxalactam and zinc ions in catalytic site over time. The 2Fo-Fc map was contoured at 1.0 σ level (carved at 1.4 Å) around moxalactam and zinc ions in catalytic site of L1 MBL. We have also modified Supplementary Figure 4 and added panel b (showing the 2Fo-Fc map contoured at 1.0 σ level (carved at 1.4 Å) around moxalactam) and panel c (showing the 2Fo-Fc polder OMIT map countered at 2.0 σ level (carved at 2 Å) around moxalactam. Polder OMIT map were calculated using Phenix (Liebschner et al. 2017) with the exclusion of bulk solvent in radius of 5Å from moxalactam, resolution factor 0.25 were used during calculation).

Q. Page 13 -- Figure 4

“a. L1 active site structure at 150 ms...”

“b. Electron density...”

“f. TR “pink” beam structures...”

Please add labels for a few of the key residues. It is also wise to include interactions between the Zn atoms and their coordinating atoms to help illustrate coordination sphere.

Response: We added labels as suggested by reviewer to a Figure 4a and depicted the coordination of zinc atoms using green lines. The figures 4b and 4f have been substituted with close view illustration of b-lactam ring cleavage, without depiction of protein residues sidechains.

Q. “g. Catalytic mechanism proposed for L1 MBL based on TR-SSX experiments.”

It would be good to indicate that these “intermediate” structures are not observed in the time-resolved atomic models and apparently happen within the 50 ms between the 100 and 150 ms time points. It would also be good to indicate if the activated water/hydroxide is either the bridging solvent or a terminal ligand to one of the Zn atoms.

Response: We edited the text as suggested by reviewer.

Q. The coordination of the solvent molecule(s), protein ligands, and substrate atoms to the Zn ions are not indicated. This makes it more difficult to figure out where the proposed nucleophilic solvent is “created” and from what direction it attacks the substrate bond to be broken. In principle, this is the type of information that could/should come from a time-resolved study. However, it appears that in these results the modest resolution and/or dynamics of the reaction cloak the solvent atoms to crystallographic analysis and yield less than obvious electron density maps.

Response: We are not sure which water molecule is activated. The likely candidate is water 423. It is present in 100 ms time point but missing in 150 ms time point, or perhaps is now a part of newly formed carboxylate. Similar water was observed in our simulation of NDM-1.

„The incoming water (for example W423) is activated and water/hydroxide attacks C-8 and forms intermediate anion(s)”

Q. In addition, the detailed mechanistic paragraphs (from the last paragraph on page 13 through middle of page 16) should include a better figure for the proposed reaction mechanism than shown in part “g”. An improved figure will also help with the discussion text, which also needs to be sharpened and a bit more focused. For instance, at the page 15 -16 interface the authors state, “A water molecule can be attracted by di-zinc center, and it becomes activated “attacking” water and it rapidly reacts with C-8 that is precisely positioned for reaction.” Here the text is very poorly worded; this matters, and an “activated water” is probably better described as a hydroxide ion (as stated a few line lower), not a water molecule. The combination of a poor mechanism figure and wandering text significantly decreases the impact of results and insights gleaned from the TR-SSX structures.

Response: Yes, we have edited reaction mechanism paragraph, and use “hydroxide ion” in place of activated water.

Q. Page 18

“...we indexed, and integrated 5191 (3374) diffraction images per data set.”

Throughout, it would be clearer to indicate the number of crystal lattices merged per dataset since some images may or may not contain more than one lattice. Despite many thousands of images with apparent diffraction spots (e.g. table of data collection and refinement stats), there are only a few hundreds of

crystal lattices actually merged. Why do so many images exhibit strong spots, but apparently fail to integrate and/or merge into the whole TR-SSX dataset?

Response: This paragraph refers to monochromatic SSX data collection, we were indexing only one crystal lattice for image using DIALS. The number of merged diffraction images decreased due to post indexing selection algorithm that uses script which reject images that possesses only low-resolution diffraction spots. In this case we removed from integration images that doesn't have spots that exceeded resolution cut-off 2.0 (2.2) Å, respectively for "no zinc" and "one zinc" (Supplementary Table S2A). Importantly the outlier rejection procedure reduces R factors of refined structures and doesn't affect completeness of the data at last resolution shell (both reported structures have 100% completeness). Due to high crystal symmetry, we obtained very good redundancy 76.5 (54.3), thus we decided to select fraction of best diffraction images for data integration.

For pink-beam TR datasets the number of crystal lattices merged per data set is quite large. We include the number of merged lattices in Table S1B. Assuming that the x-ray beam strikes entire volume of crystal for 100 ms and 150 ms time points the number of unit cells is respectively $\sim 1 \times 10^9$ and 0.5×10^9 , respectively. This gives rise to excellent diffraction patterns as shown in Fig. S2C.

Q. Page 18

"... polychromatic x-rays (1.02-1.18 Å wavelength range) were focused using Kirkpatrick–Baez mirrors..."

Please also add values in eV too, and indicate the % band pass for these pink beam studies.

Response: The bandpass was 5% or 600eV (FWHM). We added this to the text.

Q. Page 20

"We used nanosecond pulses from an OPOTEK Opolette 355 II HE laser, focused to 100 x 80 μm spot."

More detail is needed for the photo-physics of the de-caging reactions. How many nanosecond pulses for what total illumination time were used? What was the optical light intensity/power for the de-caging reaction? What is the quantum efficiency given the illumination and sample conditions?

Response: We used only one ns laser pulse per crystal for de-caging. Reaction was triggered with 5.5 mJ/mm² laser power density. The [Zn(XDPAdCage)]⁺ has a quantum efficiency of 27% (Basa et al. 2019). Using 365 nm wavelength of the 100x80 μm focused laser we estimate that in single pulse we decompose 2.07×10^{13} caged molecules.

Fig. In plane geometry of the UV-laser used during experiment.

Q. Page 24

“Supplementary Figure S1. Photolysis of [Zn(XDPAdCage)]⁺ with 347 nm UV light in solution triggers cleavage of meropenem in a presence of L1 from *S. maltophilia*. Prior experiment zinc ions were removed from L1 by dialysis against buffer B containing EDTA.”

More information is needed, please. Is this analysis for enzyme in solution or in a single crystal, or a crystal slurry? How much EDTA for how long and at what temperature, pH? It is very difficult to tell if these spectra differ from spectra shown in ref 32 (*J. Am. Chem. Soc.* 141, 12100–12108 (2019)). The text suggests that there is a change in the optical spectrum upon hydrolysis of moxalactam; however the caged Zn compound and photoproducts are also coloured and contribute to the observed spectra. It would be good to help the reader deconvolute the optical spectra.

Response: We have added spectra for meropenem and [Zn(XDPAdCage)]⁺ alone to illustrate that cage spectrum has a local maximum of absorption at 245 nm. However, the deconvolution of spectra shows that presence of [Zn(XDPAdCage)]⁺ do not significantly overlaid with spectrum of uncleaved meropenem. We included supplementary figure S1b.

Q. Page 24

“Final concentration of moxalactam was 6.58 mM and the concentration of [Zn(XDPAdCage)]⁺ was 3.29 mM.”

Please provide the reader with an estimate for the enzyme concentration (and the active site metal concentration) within the crystals. The reader may note that the substrate concentration is twice that of the Zn, but each active site binds two Zn atoms and only one antibiotic. Please provide a K_d and/or K_m for L1 and moxalactam. One may assume that the crystals are several mM enzyme, and therefore wonder if this is enough metal to fully occupy the active sites throughout the crystal. The reader will also wonder what the quantum efficiency of the de-caging process is; please provide a brief summary of the photo-physics characteristics.

Response: We have provided in a text concentration of protein used for batch crystallization which was 47.9 mg/ml (1.66 mM), after mixing with crystallization buffer concentration of enzyme decreased to 0.83 mM. Crystals of L1 MBL were washed three times, that procedure removes all enzyme from the solution, and we assume that only enzyme molecules in SSX data collection are present in crystals. Crystals were harvested by centrifuging batch samples and transferred to a nylon mesh for SSX data collection. This makes the indirect calculation of enzyme concentration dependent on many factors. Possibly the concentration of L1 MBL during the TR-SSX experiment could be directly measured by dissolving crystals in 5% DMSO and subsequent absorption measurement. However, we don't have the materials to repeat this experiment. The major point is that the concentration of the cage during the experiment is sufficient to partially saturate the active site of L1 MBL, on average we obtained occupancy of 59.5 % and 74.7 %, respectively for Zn1 and Zn2. In comparison occupancy of refined reference data that were not treated with EDTA is 82% for Zn1, and 80% for Zn2. Moreover, [Zn(XDPAdCage)]⁺ has improved photophysical properties in comparison with previously available zinc cages. The [Zn(XDPAdCage)]⁺ has a quantum yield of 27% and binds zinc ions with a high affinity 4.6 pM (Basa et al. 2019).

Q. Page 25

“Supplementary Movie S1. Snapshots of moxalactam cleavage by L1 from *S. maltophilia* captured by TR-SSX (20 – 4000 ms).”

The reviewer did not have access to Movie S1. Is the legend detailed enough so that the reader understands what is being shown? Are the transitions between time-points molecular morphs or does it show authentic steps without morphs between them. Is there electron density shown, and if so, then to what resolution and what type of maps...

Response: We included Movie S1 in the revised manuscript. The Movie S1 shows authentic steps at 10 time points. We added file with evolution of electron density (b). For clear view there is no electron density included. The figure legend explains sufficiently what is being presented. Supplementary Movie S1. Snapshots of moxalactam cleavage by L1 from *S. maltophilia* captured by TR-SSX (20 – 4000 ms). Protein is in yellow, zinc ions in magenta, water molecules are labelled red, moxalactam is in stick representation with carbon atoms in green prior β -lactam cleavage, orange right after cleavage (150 ms) and blue in the final product during conformational adjustments. **a.** conformational changes in active site moiety during moxalactam cleavage **b.** The 2Fo-Fc map contoured at 1.0 σ level (carved at 1.4 Å) around moxalactam, maps were depicted as a light-blue mesh. The Fo-Fc maps were calculated globally for each structure and depicted as green for positive and red for negative electron density (at 3.2 σ and - 3.2 σ level, respectively).

Q. Page 26

“Supplementary Figure S1. Photolysis of [Zn(XDPAdCage)]⁺ with 347 nm UV light in solution triggers cleavage of meropenem in a presence of L1 from *S. maltophilia*. Prior experiment zinc ions were removed from L1 by dialysis against buffer B containing EDTA.”

This poorly worded legend also needs significantly more detail. It is not at all clear what the figure shows. Does “NO” stand for a dark sample/control? What is /are the concentration(s) of the [Zn(XDPAdCage)]⁺ used in these experiments, temperature of the reaction?, illumination power?, other reagents in the crystal slurry or solution(s)?, How much starting meropenem is in the reactions? How much meropenem is cleaved during this time? Which lambda max goes with which species? How does this compare to Figure 4 in ref 32 (J. Am. Chem. Soc. 141, 12100–12108 (2019)) – perhaps add

another panel to show this comparison?

Response:

For this experiment we used meropenem instead moxalactam due to high absorption spectrum of meropenem, which is decreased during cleavage of β -lactam ring by β -lactamases (Brem et al. 2015). What is known the L1 MBL is highly processive enzymes, therefore we used excess of substrate over the enzyme and $[\text{Zn}(\text{XDPAdCage})]^+$, thus observed spectrum is dominated by absorption of meropenem as shown on Supplementary Figure 1. For the experiment we used 365 nm handheld UV-lamp that doesn't provide control of illumination energy for specific area. However, the concept of the experiment was to check $[\text{Zn}(\text{XDPAdCage})]^+$ decaging in crystallization buffer conditions before beam time at synchrotron. The caged compound has been decomposed in solution used for crystal handling, therefore we were convinced that we could start reaction of β -lactam cleavage in crystals of L1 MBL. Observed absorption spectrum of $[\text{Zn}(\text{XDPAdCage})]^+$ have local maximum at 347 nm as reported by Basa et al. 2019. More complex biochemical characteristic of $[\text{Zn}(\text{XDPAdCage})]^+$ could be performed in solution, but here we showed basic experiment that allowed us to work on advanced time-resolved crystallography system at 14-ID beamline.

Brem, J., Struwe, W. B., Rydzik, A. M., Tarhonskaya, H., Pfeffer, I., Flashman, E., Van Berkel, S. S., Spencer, J., Claridge, T. D. W., McDonough, M. A., Benesch, J. L. P., & Schofield, C. J. (2015). Studying the active-site loop movement of the São Paulo metallo- β -lactamase-1. *Chemical Science*, 6(2), 956–963. <https://doi.org/10.1039/C4SC01752H>

Supplementary Figure S1. Photolysis of $[\text{Zn}(\text{XDPAdCage})]^+$ with 347 nm UV light in solution triggers cleavage of meropenem in a presence of L1 from *S. maltophilia*. **a.** Activity of L1 MBL against meropenem. Prior experiment zinc ions were removed from L1 by dialysis against buffer B containing EDTA. For the measurements we used 500 μM of meropenem with 0.1 μM of L1 MBL. **b.** Control spectra depicting absorption of meropenem and $[\text{Zn}(\text{XDPAdCage})]^+$ without L1 MBL.

Q. Page 29

“Supplementary Figure S4. Evolution of electron density in the L1 MBL active site during TR-SSX experiments.”

More information is needed for the legends. For example, are these 2 Fo-Fc maps carved around residues of interest and contoured at 1 sigma (and to what resolution)? Are these typical omit maps, or are they Polder omit maps?

Response: We have updated legends.

Q. Page 30

“Supplementary Figure S5. ... (red – high B-factor; dark blue – low B-factor).”
Please be quantitative, provide the values

Response: As requested by reviewer, we updated Supplementary Fig. S5 and Table S6.

Q. Page 31

“Supplementary Table S1A. SSX data collection and processing statistic of L1 β -lactamase crystals.”
The reader will wonder what are the estimated average x-ray doses to the crystals (expressed in Gy)

used for these studies; therefore, please provide this information.

Response: We have updated Supplementary Table S1A and S1B with the estimated average x-ray doses to the crystals (expressed in Gy).

Q. What are the average B-values for the antibiotic and the Zn atoms in the TR-SSX datasets. Should the reader assume 100% occupancy for these ligands in each timepoint dataset?

Response: We have added information regarding the average diffraction weighted dose used for the determination of presented crystal structures of L1 MBL. For calculation of the x-ray dose delivered to crystals we used RADDPOSE-3D v4.0.1011 (Zeldin et al. 2013). The photon flux on the 19-ID beamline was 3×10^{12} photons per second, the pink beam at the 14-ID beamline at BioCARS generates 1×10^{20} photons per second. However, higher photon flux of the pink beam is compensated by shorter crystal exposition. Therefore, the average diffraction weighted doses for all structures determined using monochromatic and polychromatic SSX are comparable, from 11.4 kGy to 33.6 kGy (Supplementary Table S1A). These values are typical for SSX data collection that is considered as low-dose method for determination of crystal structure.

Moreover, we have re-refined 500 ms data point and re-deposited this structure to PDB, the Table S2B was updated. Because distances change a bit after refinement, we have updated supplementary Tables S3, S4 and S5.

REVIEWER COMMENTS

Reviewer #1 (Remarks to the Author):

The revised manuscript has included new information, such as the Supplemental Figure Movie S1 and Supplemental Figure S2, to better explain their experimental setup and structural findings. These revisions are well noted and much appreciated.

Here below are some major concerns and questions regarding the revised manuscript.

Q1. It is best if the panels (a-c) of Supplemental Figure S2 are added to Figure 1 to better explain the time resolved synchrotron light source (TR-SSX) system used in this study. Given that the TR-SSX is the crucial technology that underpins all the subsequent crystallographic analysis in the manuscript, it is very important that this TR-SSX system be clearly explained to the readers.

Q2. Figure 2 should be expanded to show comparison of L1 active site to that for other MBLs. This is to support the authors' statement in the abstract "Mechanistic details captured here should be generalized to other MBLs". Such generalization is only appropriate if the active site among MBLs is highly conserved. Also the structure in Figure 2 should also be compared to L1 and MBLs in complex with beta lactam substrates and inhibitors other than moxalactam. Panel (b) in Figure 2 is not necessary as the tetrameric state of L1 is already shown in Supplemental Figure S2.

Q3. Figure 3b and Supplemental Figure S5a are identical except for the Fo-Fc map in S5a. However, there were only a few small pieces of densities from the Fo-Fc map scattered around the active site and their functional significance is not clearly stated. Panels S5b and S5c in Supplemental Figure S5 look redundant as they are both Fo-Fc maps, just calculated using two different software.

Q4. The coloring scheme for the two zinc ions was not clarified in several figures, including in Figure 3c and Figure 4f for the three structures (100ms, 150ms and 2000ms).

Q5. The "Results and Discussions" section should be reorganized into subsections to highlight the key findings of the manuscript one by one. The current version shows one long section and drifted from one intermediate state to another without clearly drawing conclusion for each point.

Q6. The novelty of this manuscript, or the lack thereof, is the biggest concern.

In response to this reviewer's initial inquiry on this issue, the author's rebuttal stated that "We clearly were able to address the essentiality of both zinc ions in substrate binding and in catalysis." However, this two-zinc mode is well-accepted knowledge for L1 type of MBLs. The author's argument of Zn₂ being more important for catalysis has been stated in previous studies as shown by references cited from their own study and other labs in the field.

The author's rebuttal also stated that "For the first time we have shown that the enzyme holds the substrate as it adapts in the catalytic site without reaction occurring..... Our data suggest more dynamic mechanism with ligand conformational changes playing much more significant role than the enzyme conformational adjustments".

Indeed the structures at 20ms, 40ms, 60ms, 80ms and 100ms showed subtle conformational changes for both the substrate moxalactam and some residues within the active site like S221. (Additional note: residues like S221 should be labeled in Figure 3b to help readers understand this critical finding). However, there is no in-depth analysis by the authors, for example, molecular dynamics (MD) simulations or QM/MM analysis to show that the enzyme helps the substrate to adapt to a catalytically suitable conformation through the time course of 20ms to 100ms so that the substrate is correctly poised to undergo hydrolysis at 150ms. Such MD and QM/MM analysis would be needed to validate the novelty of the ligand conformational changes reported.

To add novelty to this manuscript, the author could consider applying the powerful TR-SSX system to study L1 hydrolyzing other substrates, like carbapenems that are more clinically relevant for antibiotic resistance. While K_m and K_{cat} could be limiting factors, it is possible some mutations in L1 may alter

enzyme kinetics in a proper manner to allow TR-SSX studies.

Overall, in this revised manuscript, the authors clarified about their experimental setup and added reference to existing literature for their results and discussions. However, findings reported here only conform to existing knowledge (both from their own previous studies and from others in the field) but lack novel mechanistic insights to inspire further studies.

Reviewer #3 (Remarks to the Author):

The authors have done a reasonably good job accommodating nearly all of the three reviewer's questions, comments, and points of clarification. These are detailed in the rebuttal letter and in the revised manuscript. In some cases, new images have been prepared to better illustrate results; for instance, some new Polder omit maps and an illustration of the crystal lattice packing. The inclusion of additional mechanistic references strengthens the overall story and impact.

Supplementary Movies S1 and S2 will benefit from annotations for the time point on each image. As it is now, the reader will have a difficult time keeping track of what image goes with what time point.

October 12, 2022

Associate Editor

Nature Communications
orcid.org/0000-0001-7393-7687

Dear Dr. Hassaan,

Please find enclosed a second revision of our manuscript entitled "***Time-Resolved β -lactam Cleavage by L1 Metallo- β -Lactamase***" by Wilamowski *et al.*

We thank the reviewers for additional suggestions to improve the manuscript. We have responded to all reviewers' comments and suggestions. We have modified figures and figure legends and movies. We modified manuscript by adding a section describing observed changes in the conformation of substrate and product. Specifically, we made the following revisions:

- Modified Figure 1 (added panels b, c and d)
- Modified Figure 2, we have added the figures with the depiction of the crystal structures (b) and active sites (c) of the representatives of B1, B2, and B3 subclasses of MBLs
- Removed panel S5a from the supplementary Figure, the information about the fitness of the model to experimental electron density based on Fo-Fc differential maps is displayed as Supplementary Movie S1b.
- Added the legend for the color scheme (left site of image) illustrating shades of pink spheres for zinc atoms displayed in Figure 3c and 4f.
- Modified panel 3b. We have moved time-point labels to the top corner of images presented in Figure 3b to get clear view of the active site of L1 MBL.
- Provided more detailed description of conformational changes of the substrate and product in the active site. This is included in the manuscript in the section chemical transformation.
- Added labels to supplementary movies.

All changes are shown in red font. The detailed response to the reviewer comments and questions is provided below.

Very truly yours,

Andrzej Joachimiak, Ph.D.
Structural Biology Center

Response to reviewers

Reviewer #1:

The revised manuscript has included new information, such as the Supplemental Figure Movie S1 and Supplemental Figure S2, to better explain their experimental setup and structural findings. These revisions are well noted and much appreciated.

Q1. It is best if the panels (a-c) of Supplemental Figure S2 are added to Figure 1 to better explain the time resolved synchrotron light source (TR-SSX) system used in this study. Given that the TR-SSX is the crucial technology that underpins all the subsequent crystallographic analysis in the manuscript, it is very important that this TR-SSX system be clearly explained to the readers.

Response: As suggested by reviewer Figure 1 has been modified to illustrate in detail the methodology of serial synchrotron crystallography used in pump-probe time-resolved studies of L1 MBL. In agreement with the reviewer's request in Figure 1, we have added panels b, c, and d that previously were present in Supplementary Figure S2. We updated the figure legend as well.

Q2. Figure 2 should be expanded to show comparison of L1 active site to that for other MBLs. This is to support the authors' statement in the abstract "Mechanistic details captured here should be generalized to other MBLs". Such generalization is only appropriate if the active site among MBLs is highly conserved. Also, the structure in Figure 2 should also be compared to L1 and MBLs in complex with beta lactam substrates and inhibitors other than moxalactam. Panel (b) in Figure 2 is not necessary as the tetrameric state of L1 is already shown in Supplemental Figure S2.

Response: We expanded Figure 2 and compare active site of L1 with other MBLs. To compare the structures of MBLs we have added the figure with the depiction of the crystal structures and active site of the representatives of B1, B2, and B3 subclasses of MBLs. We superimposed L1 MBL crystal structure determined at room temp using SSX (PDB entry 7L52 - magenta) with structures of MBL representatives: B1 subclass NDM-1 from *K. pneumoniae* (PDB entry 6TWT - cyan), B2 subclass mono-zinc MBL SFH-I from *Serratia fonticola* (PDB entry 3SD9 - yellow), B3 subclass FEZ-1 MBL from *Legionella gormanii* (PDB entry 5WCK - grey).

Q3. Figure 3b and Supplemental Figure S5a are identical except for the Fo-Fc map in S5a. However, there were only a few small pieces of densities from the Fo-Fc map scattered around the active site and their functional significance is not clearly stated. Panels S5b and S5c in Supplemental Figure S5 look redundant as they are both Fo-Fc maps, just calculated using two different software.

Response: As requested by the reviewer and to avoid redundancy, we have removed from supplementary figures panel S5a, the information about the fitness of the model to experimental electron density based on Fo-Fc differential maps is displayed as Supplementary Movie S1b. We have left only polder OMIT maps and removed panel S5b which was a close-up view of the L1 MBL active site around the β -lactam ring, similar information contains the data presented in Figures 3b and 4f in the main manuscript. The polder OMIT maps were combined with Supplementary Figure S4. The tetrameric state of L1 depicted in Figure 2b was moved to supplementary Figure S2a which contains information presented in S2b about the crystal packing of L1 MBL crystals used in the TR-SSX experiment.

Q4. The coloring scheme for the two zinc ions was not clarified in several figures, including in Figure 3c and Figure 4f for the three structures (100ms, 150ms and 2000ms).

Response: We added the legend for the color scheme (left site of image) illustrating shades of pink spheres for zinc atoms displayed in Figure 3c and 4.

Q5. The “Results and Discussions” section should be reorganized into subsections to highlight the key findings of the manuscript one by one. The current version shows one long section and drifted from one intermediate state to another without clearly drawing conclusion for each point.

Response: We reorganized the Conclusion section to indicate key findings. Specifically, we emphasize:

- visualization of progression of binding metal ions and substrate,
- no reaction occurring during first 100 ms of process,
- recognition of β -lactam by metal scaffold,
- interaction of other moieties of the ligand with protein are secondary and this explains enzyme promiscuity,
- activation of water molecule to hydroxyl (no bridging water molecule bound to zinc scaffold is present prior reaction),
- change in inter-atom distances during chemical transformation,
- point to oxygen in newly formed carboxylate originating from attacking hydroxide,
- unexpected rotation of oxazine ring during reaction,
- conformation of product after reaction is similar to substrate.

Q6. The novelty of this manuscript, or the lack thereof, is the biggest concern.

Response: We believe our data are unique, yes they confirmed some previous observation but showed completely unexpected view of the active site. Moreover, the substrate is bound for over 100 ms without β -lactam cleavage and the reaction occurs without bridging water bound to zinc ions prior reaction. The substrate and product undergo conformational changes that have not been predicted, including oxazine ring rotation.

The reviewer suggests to conduct “in-depth analysis by the authors, for example, molecular dynamics (MD) simulations or QM/MM analysis to show that the enzyme helps the substrate to adapt to a catalytically suitable conformation through the time course of 20ms to 100ms so that the substrate is correctly poised to undergo hydrolysis at 150ms. Such MD and QM/MM analysis would be needed to validate the novelty of the ligand conformational changes reported.”

We have done such analysis for NDM-1 (Kim et al, 2013), however, to perform this for all data points of our TR SSX would be very significant effort both in preparation all the simulation parameters for observed states and defining correct computational trajectories as well. Recent publications on QM/MM simulation on L1 from Mulholland lab (Twidale et al. J Chem Inf Model. 2021 PMID: 34637298) and by Krivitskaya and Khrenova, J Chem Inf Model. 2022 PMID: 35758922, shows that this is not trivial task requiring significant effort and resources. Therefore, we believe including such computational experiments for this manuscript is significant project by itself and is beyond the scope of our experimental work and it will unnecessarily delay publication.

The TR-SSX is rather complex experiment to perform and that is why it has been only reported at much narrower scope for β -lactamases. Our work is experimental and provides basis for theoretical calculations both using MD and QM/MM. By depositing and releasing structures to biology community we make our data open to other researchers to study ligand binding, interaction and catalysis.

The reviewer also suggested to apply this “powerful TR-SSX system to study L1 hydrolyzing other substrates, like carbapenems that are more clinically relevant for antibiotic resistance.”

We fully agree with this statement and in fact we are planning to apply to FELs facilities to perform TR SFX (Serial Femtosecond Crystallography) on different substrates at finer time slices with premise to capture attacking hydroxide and observe differences in reaction for different substrates. We have published recently structures of complexes of L1 with several ligands (Kim et al. Protein Sci. 2020 PMID: 31846104). In this work we reported binding of moxalactam, penicillin G, captopril, imipenem and meropenem to L1 MBL, therefore addressing some reviewer points.

MD and QM/MM can provide more plausible analyses for molecular dynamics particularly when experimental data are not available. However, it still requires significant effort, and the results need to be carefully validated, with a caveat whether MD and QM/MM can validate experiments or rather experiments can validate the calculations. In our recent work on other drug discovery projects (particularly SARS-CoV-2 drug targets) we have collaborated with several computational groups trying to predict binding of small ligands to active site with rather very low success rate).

AS suggested by reviewer we have modified panel 3b, indeed labels with the time point of the reaction depicted on the bottom corner covered the S221 residue at the active site of the L1 MBL. Thus, we have moved time-point labels to the top corner of images presented in Figure 3b to get clear view of the active site of L1 MBL.

As requested by reviewer we provide more detailed description of conformational changes of the substrate and product in the active site. This paragraph is included in the manuscript in the section chemical transformation (see below).

“During time leading to the bond breakage (between 20 and 100 ms) there is no significant distance changes between ligand and protein as indicated in Table S4 and depicted on Figures 4c, S4d. This is consistent with the idea that the substrate molecule fluctuates slightly prior the catalytic event. This also includes oxazine as the same ring configuration and bond distances are always maintained in the structures in all time points. After the bond scission occurs between C-8 and N-5 atoms there are some significant distance changes, and the product settles down by binding to two zinc ions and stays bound till 4000 ms time point. Unexpectedly during catalysis, the oxazine ring rotates counterclockwise (~20 degrees) along the C6-C7 bond as scissile bond gets broken between 100 and 150 ms (Fig. S4d, e). The oxazine ring is also flatter at 150 ms than at other time points. Then the ring rotates back clockwise from 500 to 4000 ms and returns to the configuration similar to that of the substrate. In the process, atoms N5 and O14 (as the C13 carboxyl turns slightly) approach Zn2 (reducing distance from 2.4 to 2.2 Å and 2.9 to 2.3 Å, respectively) (Fig. S4e). The product is then stabilized and stays bound to both zinc ions. Right after the scission event when a hydroxyl molecule attacks carbonyl C8, newly formed C8 carboxyl moiety is well visible in electron density (Fig. 4f). The O10 atom most likely comes from the attacking hydroxyl molecule (not from the carbonyl O9) as it is located closer to the Zn1. Interestingly all possible residues directly involved in catalysis, for example,

activating solvent molecule and providing a proton (Zn2, His and Asp) are located near Zn1. Because of the resolution of the structures, it is not possible to address possible molecules tautomeric states during the catalysis.”

We also updated with figures S4d and S4e, as shown below.

Figure S4. Comparison of moxalactam movement in the L1 MBL active site observed from 20-100 ms after UV-pulse (before reaction). **e.** Moxalactam conformation changes after reaction, time frames 150-4000 ms.

Reviewer #3

The authors have done a reasonably good job accommodating nearly all of the three reviewer's questions, comments, and points of clarification. These are detailed in the rebuttal letter and in the revised manuscript. In some cases, new images have been prepared to better illustrate results; for instance, some new polder omit maps and an illustration of the crystal lattice packing. The inclusion of additional mechanistic references strengthens the overall story and impact.

Q1. Supplementary Movies S1 and S2 will benefit from annotations for the time point on each image. As it is now, the reader will have a difficult time keeping track of what image goes with what time point.

Response: We appreciate the contribution of the reviewer in improving our manuscript during the first round of revision. The detailed questions raised previously allowed us to perform more advanced description of the studied crystallographic system in the revised manuscript. According to one left suggestion, we have added time-point labels on images presented in Supplementary Movies S1a and S1b.